**RESEARCH**

# A global screening identifies chromatin-enriched RNA-binding proteins and the transcriptional regulatory activity of QKI5 during monocytic differentiation

Yue Ren[1,2†], Yue Huo[1,2†], Weiqian Li[1,2†], Manman He[1,2], Siqi Liu[1,2], Jiabin Yang[1,2], Hongmei Zhao[2,3], Lingjie Xu[4], Yuehong Guo[1,2], Yanmin Si[1,2], Hualu Zhao[1,2], Shuan Rao[5], Jing Wang[2,3], Yanni Ma[1,2], Xiaoshuang Wang[1,2*], Jia Yu[1,2,6*] and Fang Wang[1,2*]

* Correspondence: cattle1131@163.com; j-yu@ibms.pumc.edu.cn; wo_wfang@hotmail.com
†Yue Ren, Yue Huo and Weiqian Li contributed equally to this work.
[1]State Key Laboratory of Medical Molecular Biology, Department of Biochemistry and Molecular Biology, Institute of Basic Medical Sciences, Chinese Academy of Medical Sciences, School of Basic Medicine Peking Union Medical College, Beijing 100005, China
Full list of author information is available at the end of the article

## Abstract

**Background:** Cellular RNA-binding proteins (RBPs) have multiple roles in post-transcriptional control, and some are shown to bind DNA. However, the global localization and the general chromatin-binding ability of RBPs are not well-characterized and remain undefined in hematopoietic cells.

**Results:** We first provide a full view of RBPs' distribution pattern in the nucleus and screen for chromatin-enriched RBPs (Che-RBPs) in different human cells. Subsequently, by generating ChIP-seq, CLIP-seq, and RNA-seq datasets and conducting combined analysis, the transcriptional regulatory potentials of certain hematopoietic Che-RBPs are predicted. From this analysis, quaking (QKI5) emerges as a potential transcriptional activator during monocytic differentiation. QKI5 is over-represented in gene promoter regions, independent of RNA or transcription factors. Furthermore, DNA-bound QKI5 activates the transcription of several critical monocytic differentiation-associated genes, including CXCL2, IL16, and PTPN6. Finally, we show that the differentiation-promoting activity of QKI5 is largely dependent on CXCL2, irrespective of its RNA-binding capacity.

**Conclusions:** Our study indicates that Che-RBPs are versatile factors that orchestrate gene expression in different cellular contexts, and identifies QKI5, a classic RBP regulating RNA processing, as a novel transcriptional activator during monocytic differentiation.

**Keywords:** RNA-binding proteins (RBPs), Transcriptional regulation, QKI5, Monocytic differentiation

## Background

Conventional RNA-binding proteins (RBPs) associate with structural motifs on RNA molecules via their well-defined RNA-binding domains. In doing so, RBPs help RNA to progress through different stages of its life, including alternative splicing, transport,

modification, editing, decay, and translation [1, 2], thereby mediating the roles of RNA in numerous biological processes and diseases. However, recent studies have indicated that some RBPs can also interact with chromatin and regulate gene transcription. One of the earliest examples of an RBP involved in transcriptional regulation came from a study in mouse embryonic stem cells, where the RBP Lin28A was reported to bind near transcription start sites (TSSs) and recruit Tet methylcytosine dioxygenase 1 (Tet1) to control gene transcription [3]. Since then, several other chromatin-interacting RBPs have been unveiled, revealing multiple possible mechanisms of action, for example: the RNA methyltransferase like 3 (METTL3), which is recruited to chromatin by transcription factor CRPBPZ to induce m$^6$A modification on associated mRNAs in a human leukemia cell line (MOLM13) [4]; heterogeneous nuclear ribonucleoprotein U (HnRNPU) which helps maintain the 3D chromatin structure with CCCTC-binding factor (CTCF) and RAD21 cohesin complex component (RAD21) through oligomerization with chromatin-associated RNAs in a human liver carcinoma cell line (HepG2) [5]; and WD repeat domain 43 (WDR43) which is recruited to promoters by noncoding/nascent RNAs to release Pol II, facilitating transcriptional elongation in embryonic stem cells (ESCs) [6]. Recent studies using chromatin immunoprecipitation sequencing (ChIP-seq) analysis of a selection of individual RBPs in K562 and HepG2 cells revealed that multiple nuclear RBPs were tightly associated with chromatins [7, 8]. Therefore, it seems that the chromatin-binding capacity of RBPs may represent a previously under-appreciated layer of gene expression regulation. What remains unclear is whether the majority of RBPs, which are yet unstudied, also have chromatin-binding capacity, and if so, what the physiological relevance of chromatin-binding RBPs is in human hematopoietic cells in particular, where RBPs are known to play key roles.

To answer these questions, we first identified all chromatin-binding RBPs in the nuclear fraction of three immortalized human cells using mass spectrometry (MS). We found that approximately 9.6% (52/544) of all annotated RBPs were commonly chromatin-enriched (Che-RBPs), and within them we defined a group of 7 hematopoiesis-related Che-RBPs: ADAR, PTBP3, KHSRP, ELAVL1, NUDT21, SETD1A, and QKI5. Given the known regulatory roles of QKI5 as a classical RBP in hematopoiesis [9–11], we next asked whether its novel chromatin-associating functions played a role during monocytic differentiation. We found that QKI5 located on genomic loci of several target genes in monocytic cells, activating their transcription among which was *CXCL2*, coding for a cytokine essential for monocytic differentiation [12–14]. Taken together, our data show that chromatin-binding is a property shared by a small but significant minority of RBPs. Among these, QKI5 may play an important role in regulating gene transcription during monocytic differentiation. These findings provide a reference for the study of other Che-RBPs, with an emphasis on the hematopoietic compartment, where they may be involved in the regulation of gene expression during cellular differentiation and other yet unknown processes.

## Results
### A comprehensive screening identifies numerous RBPs positioning on chromatin
Numerous RBPs localize in the nucleus where they participate in various RNA metabolism processes, such as RNA capping, splicing, and transporting [15]. Alongside,

individual chromatin-binding RBPs have been identified in the nuclei of specific cell types [8]. However, a comprehensive screening to identify all chromatin-associated RBPs in a given cell type has not been carried out. To address this, we extracted nuclei from the human hematopoietic cell lines K562 (chronic myelogenous leukemia) and THP-1 (acute monocytic leukemia), as well as HEK293T cells (human embryonic kidney expressing mutated SV40 large T antigen, also known as 293T cells), and separated the soluble components (soluble nuclear extract, SNE) from the chromatin pellets (chromatin-pellet extract, CPE) (Fig. 1a, Additional file 1: Figure S1a). We then used mass spectrometry to identify and quantify RBPs and other nuclear proteins such as transcription factors (TFs) in SNE and CPE (Fig. 1a, Additional file 1: Figure S1b). To enable us to distinguish RBPs from other nuclear proteins, we employed the RBPDB and ATtRACT databases [16, 17] to generate an RBP library (Additional file 1: Figure S1c and Additional file 2: Table S1); and JASPAR and Hoocomoco11 [18, 19] databases to construct a library for TFs identification (Additional file 1: Figure S1c and Additional file 2: Table S1). Together, by comparing our sub-nuclear mass spectrometry results with the RBP and TF libraries we were able to identify and plot the relative distribution of nuclear RBPs and TFs in the CPE or SNE fractions in the 3 cell lines. Overall, we identified 257 nuclear RBPs across the tested cell lines, of which 50.2% (129/257) were more associated with the CPE than the SNE (defined as "chromatin-enriched RBPs," abbreviated as "Che-RBPs") (Fig. 1b–d, Additional file 1: Figure S1d and Additional file 3: Table S2). In total, we identified 92 Che-RBPs in THP-1 cells (Fig. 1b), 99 in K562 cells (Fig. 1c), and 79 in 293T cells (Fig. 1d), with 52 of these Che-RBPs being present in all three cell lines (Additional file 1: Figure S1d-e and Additional file 3: Table S2), accounting for approximately 13.8% of all detected chromatin-enriched proteins (Fig. 1e, Additional file 4: Table S3) and 9.6% of all annotated RBPs (Fig. 1f). We also identified several well-known TFs on the chromatin of the cell lines (Additional file 1: Figure S1f, g and Additional file 3: Table S2). When we compared the 52 commonly detected Che-RBPs from our screening with reference databases of dsDNA-binding proteins [20], we found that 59.6% (31/52) of our common Che-RBPs overlapped with previously identified dsDNA-binding proteins [20] (Fig. 1f).

We then employed gene ontology (GO) analysis to predict the molecular functions of the 52 common Che-RBPs. This showed that these Che-RBPs expressed in the three cell lines were mainly enriched in GO terms "RNA splicing and RNA transport" (Fig. 1g, Additional file 4: Table S3). Moreover, Che-RBPs with RNA recognition motifs (RRM), CCCH zinc finger domains, or KH domains tended to enrich on chromatin (Fig. 1g). To verify the chromatin association of the top six Che-RBPs that were most abundant in CPE fractions, we performed immunoblots using the CPE fractions from different cell lines with or without RNase A treatment (Additional file 1: Figure S1h) to assess whether their chromatin associations were RNA-dependent, alongside the SNE fraction. Histone H3 and small nuclear ribonucleoprotein (SNRNP70) served as positive controls for CPE and SNE components, respectively (Fig. 1h). The results confirmed that all tested Che-RBPs displayed a chromatin-associating feature, supportive of our screening data; however, whether their associations with chromatin were RNA-dependent varied among cell lines. Serine/threonine-protein phosphatase 1 regulatory subunit 10 (PPP1R10) bound to chromatin more strongly when RNA was absent in THP-1 cells and 293T cells, indicating that RNA might obstruct its interaction with

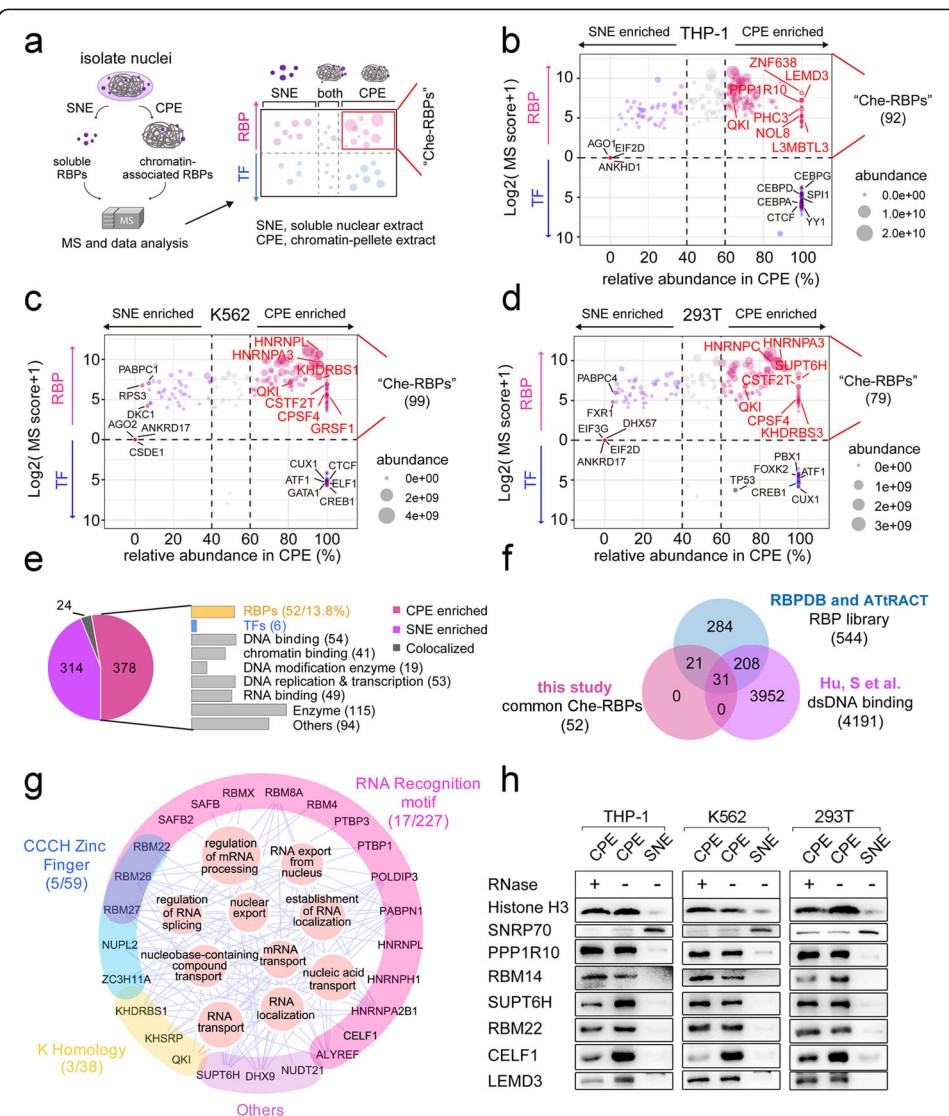

**Fig. 1** Proteomic screening for Che-RBPs in three cell lines. **a** Schematic diagram showing the major steps of mass spectrometry (MS) screening and data analysis: proteins were collected from the soluble nuclear extract (SNE) and chromatin-pellet extract (CPE) fractions separately and identified by MS data analysis. **b–d** Sub-nuclear distribution of RBPs and TFs in THP-1 (**b**), K562 (**c**), and 293T (**d**) cells. The x-axis indicates the relative abundance of RBPs and TFs in the CPE fraction (protein area value in the CPE fraction/area value in the sum of SNE and CPE fractions). The y-axis shows the log2 (MS score + 1) of RBPs and TFs in the CPE fraction. The bubble size indicates the protein area value in the CPE fraction, and bubble color represents the different sub-nuclear location of RBPs and TFs. The MS score reflects reliability of identification. And the area value represents the relative abundance of protein. **e** Molecule function categories of common proteins located in the CPE fraction based on the DAVID database among three cell lines. Some proteins belonged to multiple categories and so were counted more than once. **f** Venn diagram showing the overlap between the common Che-RBPs expressed across three cell lines identified by the MS screening, known RBPs from RBP libraries (RBPDB and ATtRACT) and dsDNA-binding proteins reported in published studies. **g** Functional annotation and domain classification of the 52 common Che-RBPs. The network represents the affiliation of RBPs to related biological processes, based on gene ontology (GO) functional enrichment analysis, and the inner circle size indicates − log10 (P value) of GO functional enrichment analysis. The color of the outer ring corresponds to the different domains present in the RBPs. **h** Immuno-blot analysis of the distribution of the common Che-RBPs that were most abundant in the CPE fractions with/without RNase treatment

chromatin. Similar results were also observed with RNA-binding motif protein 14 (RBM14) in THP-1 cells, transcription elongation factor SPT6 (SUPT6H), pre-mRNA-splicing factor RBM22 (RBM22), and inner nuclear membrane protein Man1 (LEMD3) in 293T cells (Fig. 1h, Additional file 1: Figure S1i). In other cases, Che-RBPs showed an RNA-mediated association to the chromatin, such as CUGBP Elav-like family member 1 (CELF1), which did not associate with chromatin when RNA was removed (Fig. 1h, Additional file 1: Figure S1i).

Overall, we conducted a comprehensive screening of all detectable chromatin-enriched RBPs and identified both common and unique Che-RBPs across three human cell lines. We found that some of these common Che-RBPs exhibited differential requirements for RNA to enable chromatin binding, which were sometimes context-dependent, indicating a high degree of flexibility in their function, perhaps suggestive of their involvement in multiple chromatin-regulatory pathways.

### The DNA/RNA association properties of hematopoietic Che-RBPs

We next sought to understand whether any of the 52 common Che-RBPs were likely to be directly involved in hematopoiesis. Also, numerous RBPs have been reported to play essential roles at the post-transcriptional level in hematopoietic regulation [21]; however, whether some of them possessing the transcriptional regulatory activity was still undefined. We first searched for known hematopoiesis-related RBPs using the Gene Ontology (GO) [22] database combined with published literature, which together identified 72 candidates (Additional file 2: Table S1) [9, 23–48]. Of these candidates, 7 also appeared on our common Che-RBPs list and were selected for further analysis: QKI5, double-stranded RNA-specific adenosine deaminase (ADAR), polypyrimidine tract-binding protein 3 (PTBP3), far upstream element-binding protein 2 (KHSRP), ELAV-like protein 1 (ELAVL1), cleavage and polyadenylation specificity factor subunit 5 (NUDT21) and histone-lysine N-methyltransferase SETD1A (SETD1A) (Fig. 2a). We confirmed their chromatin associations by immunoblots in the three cell lines, as before, and found that all the hematopoietic Che-RBPs (hChe-RBPs) localized on chromatin, but showed variable RNA dependency (Fig. 2b, Additional file 1: Figure S2a).

To investigate the possible regulatory functions of these hChe-RBPs, we generated ChIP-seq and CLIP-seq data from THP-1 cells (Additional file 1: Figure S2b, c, d) [8, 49–51]. The ChIP-seq analysis revealed that each hChe-RBP had a unique distribution pattern on the genome, yet all tended to bind to protein coding gene regions, while PTBP3 showed binding preference to lncRNA genes as well (Fig. 2c, left panel, see "Methods" for detail), and most of them (except for PTBP3 and ELAVL1) were over-represented in the promoter regions compared with the average promoter abundance in the human genome (Fig. 2d, upper panel). Additionally, CLIP-seq results indicated that hChe-RBPs preferred to bind to protein-coding RNA transcripts, with NUDT21 also showing a tendency to bind small RNA transcripts (Fig. 2c, right panel, see "Methods" for detail). However, unlike their association with chromatin, hChe-RBPs displayed more varied distribution patterns along RNA transcripts, with QKI5, PTBP3, NUDT21, and SETD1A tending to bind exons (Fig. 2d, lower panel). This was in contrast to previous reports on QKI5, where this RBP has been reported to regulate pre-mRNA splicing and locate on intronic regions of RNA in 293T cells [52, 53]; similarly,

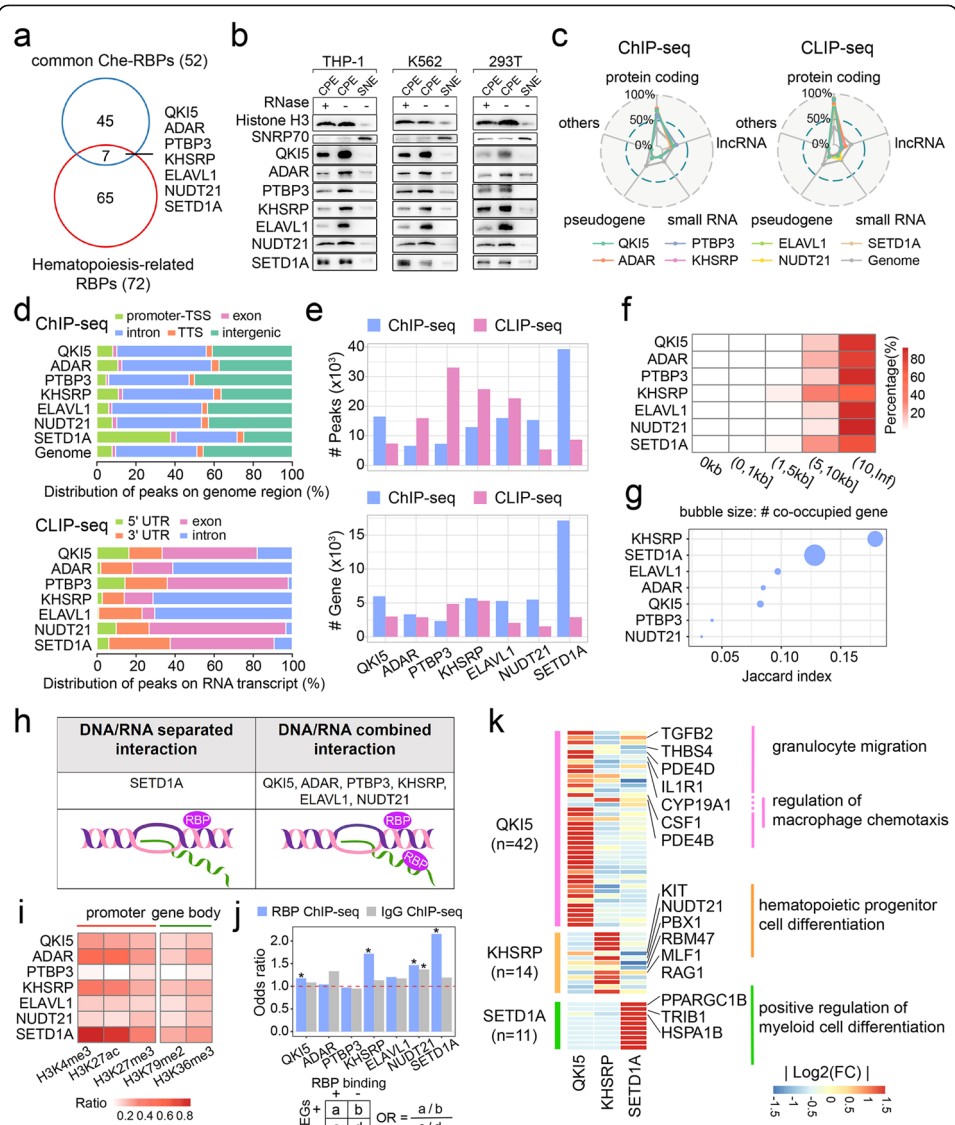

**Fig. 2** Large-scale sequencing of shared Che-RBPs. **a** Venn diagram showing the intersection between the common Che-RBPs and known hematopoiesis-related RBPs. Hematopoiesis-related RBPs were identified by literature and Gene Ontology database. **b** Immuno-blot validation of the distribution of hematopoietic Che-RBPs (hChe-RBPs) in the SNE and CPE fractions with/without RNase treatment. Immuno-blots of Histone H3 and SNRP70 were the same images from Fig. 1h since the same batch of SNE and CPE samples were used. **c** hChe-RBPs' distribution on different gene types revealed by ChIP-seq (left panel) and CLIP-seq (right panel) datasets. **d** Enrichment of hChe-RBPs on different genomic regions. Upper panel: hChe-RBPs' ChIP-seq signal distribution with human genomic intrinsic constitution as comparison. Lower panel: hChe-RBPs' CLIP-seq signal distribution. **e** Comparison of occupied peak numbers (upper panel) and gene numbers (lower panel) between ChIP-seq and CLIP-seq datasets of indicated hChe-RBPs. **f** Heatmap showing the distribution ratios of CLIP-seq peaks neighboring ChIP-seq peaks of each hChe-RBP in different distance ranges on genome. **g** Comparison of hChe-RBP co-occupied genes from ChIP-seq and CLIP-seq datasets. The x-axis shows the Jaccard index of each hChe-RBP's ChIP-seq and CLIP-seq occupied genes, with bubble size indicating co-occupied gene number. **h** Schematic diagram representing different interaction modes of hChe-RBPs to chromatin. **i** Heatmap presenting the occupation ratio of histone markers' ChIP signals colocalized with hChe-RBPs' ChIP peaks at promoter and gene body regions, respectively. **j** Upper panel: Odds ratio of differentially expressed genes (DEGs) determined by RNA-seq upon hChe-RBP knocking down on hChe-RBP-occupied versus non-occupied genes (*P value < 0.05, double-tail Fisher's exact test). The odds ratio of IgG ChIP-seq dataset versus RNA-seq datasets upon hChe-RBPs knocking down was used as a randomized control. Lower panel: Formula of odds ratio. **k** GO functional enrichment analysis of hChe-RBP-occupied DEGs with scaled absolute value of log2FC

NUDT21 was reported to conduct 3′ end processing and was mainly located in introns in 293T cells [54, 55], but here presented with dominant exon-binding features. In the case of ADAR, KHSRP, and ELAVL1, our data showed that they were mostly enriched on the introns, which were consistent with previous works in different cell types: ADAR has been reported to bind dsRNA overlapping with *Alu* elements, and conducting A-I editing in gene intron regions [56, 57], while KHSRP was shown to regulate intronic splicing and mRNA decay in the 3′UTR, exhibiting clear 3′UTR and intron-binding preference in HepG2 cells and K562 cells [58, 59], and ELAVL was reported to bind AU-rich elements in the 3′UTR and presented an intron-enriched binding pattern in 293T cells [8]. The enrichment level of each hChe-RBP also differed among different types of genes at DNA or RNA level (Additional file 1: Figure S2e). For example, QKI5 tended to accumulate on genomic regions of both lncRNA and small RNA but only showed distinct enrichment on RNA transcripts of small RNA. NUDT21 exhibited high binding intensity on RNA transcripts of small RNA but showed no gene type-specific binding tendency on genome. Moreover, besides distribution pattern analysis of gene types, we also analyzed the hChe-RBPs' distributions on different types of protein-coding genes at DNA or RNA level. We divided protein-coding genes into house-keeping (HK) and cell-type-specific genes (SP) in THP-1 cells acquired from the human protein atlas database (https://www.proteinatlas.org/) (see "Methods" for detail). For each hChe-RBP, we calculated the ratio of HK and SP genes in the RBP-bound genes at DNA or RNA level and took the ratio of HK or SP genes of THP-1 cells in overall coding genes as the reference. The binding tendency was determined by comparing the proportion of HK or SP genes in each hChe-RBP's bound genes with the reference ratio. ADAR, KHSRP, ELAVL1, and SETD1A preferred to bind genomic regions of HK genes while none of the 7 hChe-RBPs showed preference to bind SP genes. Meanwhile, all the hChe-RBPs presented binding tendency to RNA transcripts of HK genes, among which ELAVL1 also showed tendency to bind SP genes' transcripts suggesting a cell-type-specific regulatory role (Additional file 1: Figure S2f).

Taken together, we have identified seven promising candidate hChe-RBPs with likely functional roles in transcriptional regulation within hematopoietic cells, based on their common chromatin occupancy across the cell lines tested here, and published literature on their association with hematopoiesis in their conventional RNA-binding capacity. These hChe-RBPs preferentially bound protein-coding regions of the genome presenting different tendencies to HK or SP genes, with diverse enrichment in different genic regions, alongside their dissimilar RNA transcript binding capacities and tendencies.

### hChe-RBPs exhibit widespread DNA- and RNA-regulatory effects in THP-1 cells

Having established the binding patterns of the hChe-RBPs across DNA and RNA targets, we next asked whether individual hChe-RBPs displayed a preference for genomic or transcriptional targets. By comparing peak numbers from the ChIP-seq and CLIP-seq results, we found that QKI5, NUDT21 and SETD1A were more enriched on genome, while others preferred to enrich on RNA (Fig. 2e, upper panel). Meanwhile the comparison of gene numbers between ChIP-seq and CLIP-seq showed that QKI5, ELAVL1, NUDT21, and SETD1A preferred to enrich on chromatin than RNA, implying their more extensive regulation of DNA targets, while PTBP3 exhibited an RNA-

enriched pattern, indicating its regulatory preference for RNA, and ADAR and KHSRP displayed similar enrichment tendencies on DNA and RNA, suggesting their balanced regulatory features (Fig. 2e, lower panel).

In order to explore the interaction between hChe-RBPs' DNA and RNA binding, we analyzed the relative positions between ChIP-seq and CLIP-seq peaks for each hChe-RBP: The overlapping rate between ChIP and CLIP peaks of each hChe-RBP was very low, with only about 0.14% ChIP peaks overlapped with CLIP peaks in average (Fig. 2f). CLIP peaks only appeared concurrently in the range of 5~10 kb or beyond 10 kb of ChIP peaks (~ 17% and 80% in average, respectively) (Fig. 2f). In this case, we could speculate that these hChe-RBPs seemingly interacted with chromatin or RNA in unrelated ways. Moreover, though only a few genes were co-occupied by hChe-RBPs at both DNA and RNA level, we also analyzed the binding gene's overlapping ratio by introducing a "Jaccard" index (Fig. 2g). This showed that KHSRP and SETD1A possessed relatively high overlapping rates between ChIP- and CLIP-target genes which accorded with the peak-overlapping analysis indicating a multi-layered regulation role of genes' expression, while PTBP3 and NUDT21 had rather low overlapping rates. Integrating the binding feature of the hChe-RBPs generated from ChIP-seq/CLIP-seq data with the results of RNase-treated subcellular fractionation, we could speculate hChe-RBPs' interaction modes as follows: (1) DNA/RNA separated, e.g., SETD1A; (2) DNA/RNA combined, e.g., NUDT21/QKI5/PTBP3/ADAR/KHSRP/ELAVL1, summarized by the schematic diagram (Fig. 2h). From ChIP/CLIP-seq comparative analysis (Fig. 2e), SETD1A bound DNA more than RNA at both peak level and gene level, and it also showed the highest DNA-binding rate among all hChe-RBPs tested (Fig. 2e), indicating its preference to interact with chromatin, which was in accordance with RNase-treated subcellular fractionation results (Fig. 2b, Additional file 1: Figure S2a) making it belong to the DNA/RNA separated interaction mode. Other hChe-RBPs' interactions with chromatin relied on RNA at different degrees exhibited the combined interaction mode.

Next we focused on the transcriptional regulatory potential of the hChe-RBPs. We first analyzed the chromatin landscape defined by several classical histone markers on hChe-RBPs' binding sites (Additional file 1: Figure S2g, Fig. 2i). Active promoter markers like H3K4me3 and H3K27ac tended to enrich on promoter regions bound by QKI5, ADAR, KHSRP, and SETD1A suggesting that they might possess transcriptional active functions, while relatively low levels of detected histone markers presented at PTBP3-bound promoter regions as well as gene body regions indicated that PTBP3 might not be involved in transcriptional regulations.

Further, to figure out whether the hChe-RBPs' chromatin-binding feature could regulate target gene expression, we used shRNAs to generate a set of THP-1 cells each knocked down for one of the hChe-RBPs (Additional file 1: Figure S2h) and conducted RNA-seq as well as recruiting published data (Additional file 10: Table S9) to assess the resultant changes in their transcriptomes. By comparing the ChIP-seq and RNA-seq results, we were able to calculate the odds ratio (OR) between hChe-RBP-binding genes and differentially expressed genes (DEGs) in the RBP knocked down THP-1 cells. We also introduced an IgG ChIP-seq dataset generated from wildtype THP-1 cells as a randomized control representing the background noise of non-specific bindings which were irrelevant to the DEGs of knockdown effects (Fig. 2j). This identified QKI5, KHSRP, and SETD1A with potential gene expression regulatory functions. When we

conducted functional enrichment analysis of the differentially regulated gene sets, we found that these 3 hChe-RBPs influenced the expression of a set of hematopoiesis-related genes in a chromatin-associating manner, with QKI5 regulating the greatest number of functional genes (Fig. 2k, Additional file 5: Table S4).

Together, by locating on DNAs and RNAs encoding distinct products, a single RBP could exert multifunctional regulation of a broader range of targets. In THP-1 cells, we found strong evidences of functional roles for 3 of the 7 hChe-RBPs, with QKI5 appearing to be particularly potent in regulating the expression of a set of hematopoietic function-related genes.

### hChe-RBP QKI5 promotes monocytic differentiation independent of its RNA-binding ability

So far, QKI5 has been reported to regulate a range of biological processes, including hematopoiesis, via different RNA-interacting pathways [9–11]. Here, we have seen that QKI5 also potently modifies the expression of a set of hematopoiesis-related genes (Fig. 2k). Therefore, we hereafter focused on understanding the role of the chromatin association of this intriguing hChe-RBP.

Having identified high levels of QKI5 in THP-1 cells, we first asked whether its expression was uniform or varied during monocytic differentiation. We found that in both CD34$^+$ hematopoietic stem/progenitor cells (HSPCs) and THP-1 cells, QKI5 accumulated during the early stage of monocytic differentiation (Additional file 1: Figure S3a).

To dissect the effects of QKI5's chromatin- versus RNA-binding capacity during monocytic differentiation, we compared the empty vector-transduced HSPCs (Ctrl/shCtrl) to HSPC's overexpressing wild-type QKI5 (QKI5) or RNA-binding domain mutant QKI5 (QKI5 M1, which loses RNA-binding ability due to a single amino acid mutation occurring at 157th valine to glutamic acid [60]), as well as to HSPCs lentivirus-transduced with shRNAs targeting QKI5 (shQKI5-1 and shQKI5-2) (Fig. 3a, Additional file 1: Figure S3b). We found that the expression of QKI5 or QKI5 M1 significantly increased the percentage of CD14$^+$/CD11b$^+$ cells in differentiating HPSC cultures at both days 13 and 19, relative to Ctrl (Fig. 3b, left panel, Additional file 1: Figure S3c), as well as significantly increasing the number of differentiated monocytes in these cultures at day 19 (Fig. 3c). Moreover, QKI5 or QKI5 M1 expression also increased the number of monocyte-derived colony formation units (CFU-Ms) generated from HSPC cultures (Fig. 3d) compared to Ctrl. Accordingly, loss-of-function experiments showed that QKI5's reduction significantly reduced the percentage of CD14$^+$/CD11b$^+$ cells in differentiating HPSC cultures at both days 13 and 19 (Fig. 3b, right panel, Additional file 1: Figure S3d), significantly decreased the proportion of differentiated monocytes at day 19 (Fig. 3e), and significantly reduced the number of CFU-Ms (Fig. 3f), compared to Ctrl. Collectively, these results demonstrated that QKI5 promoted HSPC monocytic differentiation independent of its RNA-binding ability.

To confirm the differentiation-promoting role of QKI5, we performed a "rescue" assay in which THP-1 cells were initially lentivirus-transduced with shQKI5 then 24 h later were transduced with Ctrl, QKI5, or QKI5 M1-overexpressing lentiviruses (Additional file 1: Figure S3e). As expected, the re-introduction of either QKI5 or QKI5 M1

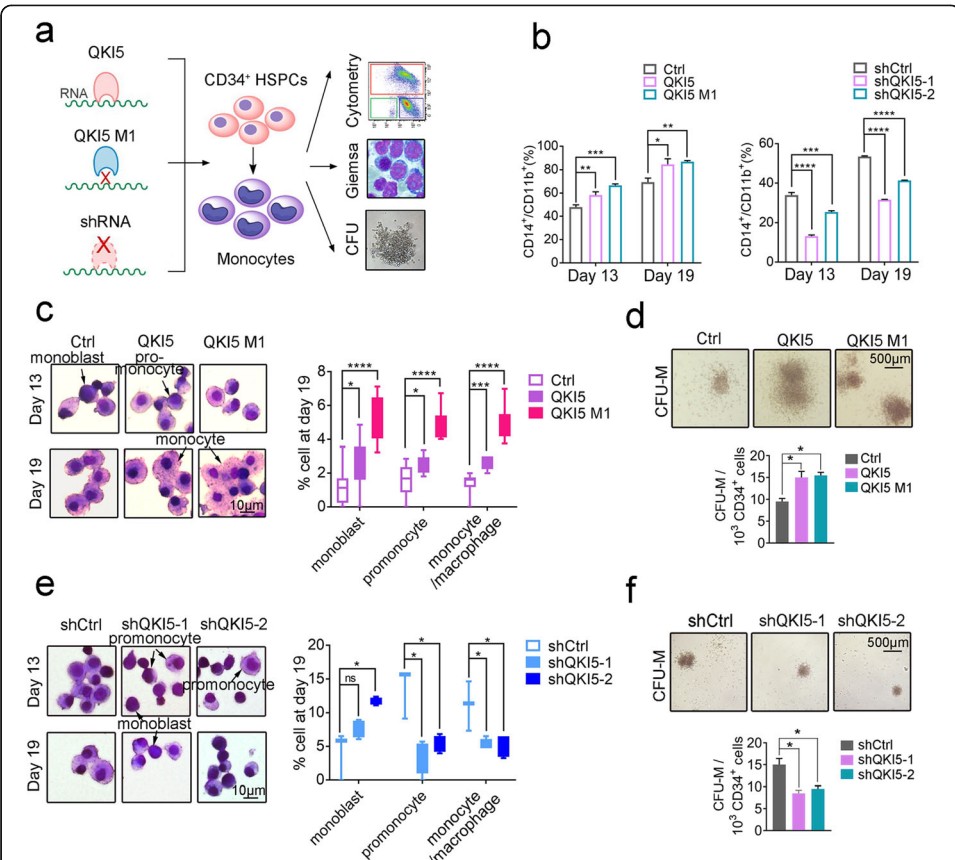

**Fig. 3** QKI5 promotes monocytic differentiation of CD34+ HSPCs. **a** Design of QKI5 functional study. **b** Left panel: Percentage of CD14+/CD11b+ cells among HSPCs within Ctrl- or QKI5/QKI5 M1-overexpressing population after 13 and 19 days of monocytic differentiation detected by flow cytometry. Right panel: Percentage of CD14+/CD11b+ cells among HSPCs within shCtrl- or QKI5 shRNAs-treated population after 13 and 19 days of monocytic differentiation detected by flow cytometry. Error bars indicate standard deviations of three biological replicates. Asterisks indicate significant differences between the indicated samples (*P value < 0.05, **P value < 0.01, ***P value < 0.001, ****P value < 0.0001, t test). **c** Left panel: Giemsa staining of Ctrl-, QKI5- and QKI5 M1-overexpressing HSPCs after 13 and 19 days of monocytic differentiation. Right panel: Average percentage of Ctrl-, QKI5-, and QKI5 M1-overexpressing cells in each cell type across the monocytic differentiation spectrum in day 19 cultures of HSPCs. Error bars indicate standard deviations around cell percentage. Asterisks indicate significant differences between the indicated samples (*P value < 0.05, ***P value < 0.001, ****P value < 0.0001, t test). **d** Colony-forming unit assay of Ctrl-, QKI5- and QKI5 M1-overexpressing HSPCs. Average frequency of colony formation per 10³ HSPCs is shown below. Error bars indicate standard deviations around counts. Asterisks indicate significant differences between the indicated samples (*P value < 0.05, t test). **e** Left panel: Giemsa staining of shCtrl-, and two QKI5 shRNA-treated HSPCs cultures after 13 and 19 days of monocytic differentiation. Right panel: Average percentage of shCtrl- and shQKI5-treated cells in each cell type across the monocytic differentiation spectrum in day 19 cultures of HSPCs. Error bars indicate standard deviations around cell percentages. Asterisks indicate significant differences between the indicated samples (*P value < 0.05, t test, ns non-significant). **f** Colony-forming unit assay of shCtrl- and shQKI5-treated HSPCs. Average frequency of colony formation per 10³ HSPCs is shown below. Error bars indicate standard deviations around counts. Asterisks indicate significant differences between the indicated samples (*P value < 0.05, t test)

successfully rescued the ability of QKI5 knocked down cells to generate CD14+/CD11b+ cells (Additional file 1: Figure S3f, g). Moreover, we confirmed that both over-expressed wild-type (QKI5) and mutant QKI5 (QKI5 M1) located on chromatin in nucleus of THP-1 cells (Additional file 1: Figure S3h), supportive of the chromatin-binding-associated function of QKI5 in monocytic differentiation.

### The wild-type and mutant QKI5 share similar gene expression profiles during monocytic differentiation

To define the influence of QKI5 on monocytic differentiation, we generated transcriptome profiles of wild-type (Ctrl), PMA-treated wild-type THP-1 cells (PMA) imitating monocytic differentiation, and QKI5- (QKI5) / QKI5 M1- overexpressing (QKI5 M1) THP-1 cells (Fig. 4a, Additional file 1: Figure S4a). We identified 2944 (QKI5-activated), 2963 (QKI5 M1-activated), and 4296 (PMA-activated) genes that were significantly more highly expressed in THP-1 cells overexpressing QKI5 or QKI5 M1, or

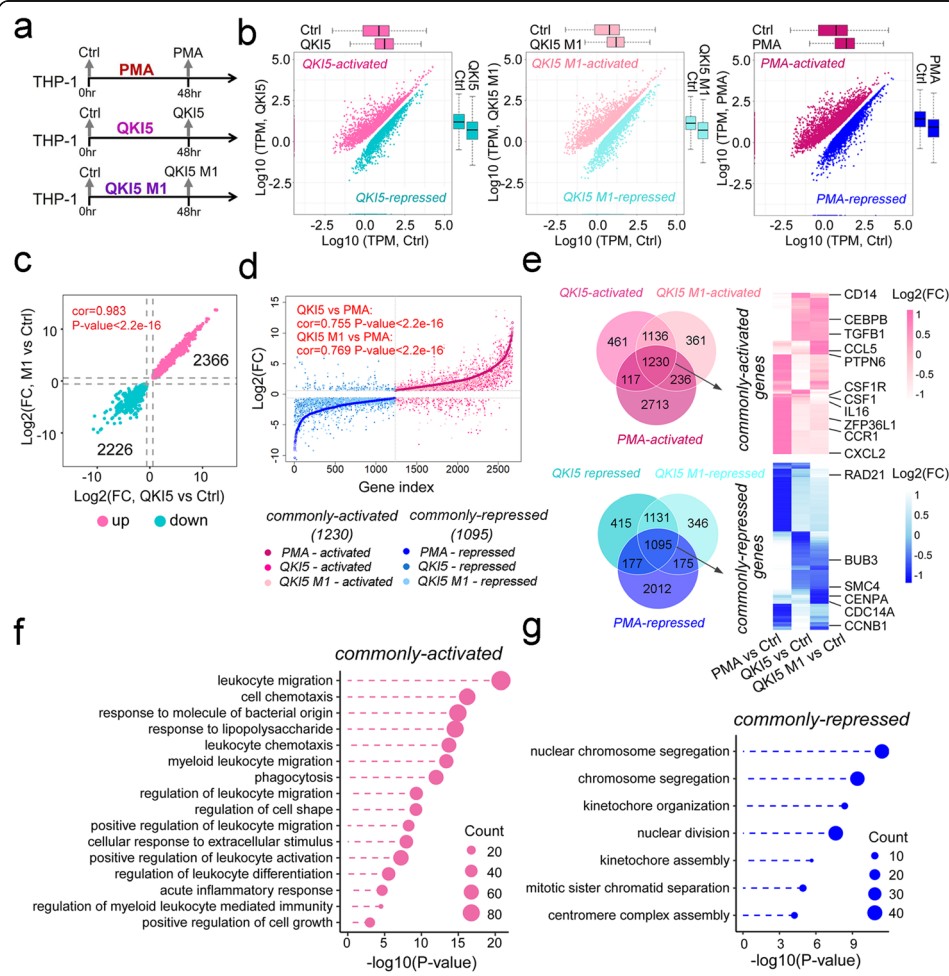

**Fig. 4** Transcriptomic effects of QKI5 protein overexpression in THP-1 cells. **a** Schematic diagram of experimental setup prior to RNA sequencing. **b** Scatterplots showing TPM values of differentially expressed genes (DEGs) between QKI5-overexpressing and Ctrl THP-1 cells (left panel), QKI5 M1-overexpressing and Ctrl THP-1 cells (middle panel), and PMA-treated (PMA 48 hr) and Ctrl THP-1 cells (right panel). **c** Scatterplots showing log2 (fold change) of up- and downregulated DEGs after QKI5 and QKI5 M1 overexpression. The correlation of gene differential expression between QKI5 regulated and QKI5 M1 regulated genes was calculated by Pearson correlation coefficient. **d** Point-by-point comparison of DEGs in QKI5- and QKI5 M1-overexpressing THP-1 cells and PMA-treated THP-1 cells. The correlations of gene differential expression between QKI5, QKI5 M1 regulated genes, and DEGs upon PMA treatment were calculated by Pearson correlation coefficient. **e** Venn diagram showing the comparison of DEGs (upper panel: commonly activated; lower panel: commonly repressed) among, QKI5-overexpressing vs. Ctrl (QKI5-activated/-repressed), QKI5 M1-overexpressing vs. Ctrl (QKI5 M1-activated/-repressed) and PMA vs. Ctrl (PMA-activated/-repressed). Heatmap on the right represents the scaled log2 (fold-change) of differential expression of commonly activated and commonly repressed genes. **f, g** GO functional enrichment analysis of commonly activated (**f**) and commonly repressed genes (**g**)

following PMA-induced differentiation, respectively, by RNA-seq (Fig. 4b). In addition, 2818, 2747, and 3459 genes were annotated as QKI5-, QKI5 M1-, and PMA-repressed genes due to their significantly lower expression following QKI5 overexpression or PMA treatment (Fig. 4b). Notably, the gene expression changes brought by overexpression of QKI5 or QKI5 M1 were highly similar (Fig. 4c), and also exhibited a strong correlation with the differentiation-associated genes affected by PMA treatment (the PMA-activated or -repressed genes) (Fig. 4d, Additional file 6: Table S5). Further, functional enrichment analysis of QKI5-, QKI5 M1-, and PMA-activated genes (defined as "commonly activated" genes) showed over-representation of monocyte-relevant terms including leukocyte migration and leukocyte chemotaxis, which contained functional genes as *monocyte differentiation antigen CD14* (*CD14*), *colony stimulating factor 1 receptor* (*CSF1R*), *colony stimulating factor 1* (*CSF1*), *interleukin 16* (*IL16*), and *C-X-C motif chemokine ligand 2* (*CXCL2*), among others (Fig. 4e, f, Additional file 1: Figure S4b and Additional file 6: Table S5). In contrast, commonly repressed genes were mainly gathered around DNA replication and cell division terms and included genes such as *structural maintenance of chromosomes 4* (*SMC4*), *centromere protein A* (*CENPA*), *cell division cycle 14A* (*CDC14A*), and *cyclin B1* (*CCNB1*) (Fig. 4e, g, Additional file 6: Table S5). Taken together, these data showed that both QKI5 and its RNA-binding-deficient mutant possessed similar effects on the transcriptome of THP-1 cells, further indicating QKI5's capacity to modulate hematopoietic gene expression independently of its RNA-binding ability during monocytic differentiation.

### QKI5 preferentially binds to activated gene promoters

Based on the ChIP-seq results, we saw that QKI5 mainly localized on the genic regions on genome (Fig. 5a) and was over-represented within the gene promoters, introns, and exons, compared with the abundance of these regions in the human genome (Fig. 5a). To investigate the potential transcriptional regulatory activity of QKI5, we undertook a further investigation of the chromatin landscape of QKI5-bound promoters. We found that the active promoter markers H3K4 tri-methylation (H3K4me3) and H3K27 acetylation (H3K27ac) aggregated around QKI5-bound promoters, as did Pol II signals, indicating a transcriptional activation role of QKI5 (Fig. 5b, Additional file 1: Figure S5a).

To verify the genomic deposition of QKI5, we selected several QKI5-occupied genes (including *C-X-C motif chemokine ligand 2* (*CXCL2*), *Mov10 RISC complex RNA helicase* (*MOV10*), *ring finger and FYVE like domain containing E3 ubiquitin protein ligase* (*RFFL*), and *uveal autoantigen with coiled-coil domains and ankyrin repeats* (*UACA*)), which also had colocalized H3K4me3 and Pol II signals (Fig. 5c). Based on the ChIP-qPCR analysis in QKI5 and QKI5 M1-overexpressing or QKI5 knockdown THP-1 cells (Fig. 5d), as expected, QKI5's enrichment on these genes significantly increased upon overexpression of either wild-type or mutant QKI5 (Fig. 5d, upper panel), and decreased when QKI5 levels were reduced by shRNAs (Fig. 5d, lower panel). Together, these data confirmed the specific occupancy of QKI5 on target genomic sites.

Furthermore, we compared the list of coding genes that were commonly activated (Fig. 4e, upper panel) and commonly repressed (Fig. 4e, lower panel) in THP-1 cells with the set of QKI5-occupied genes revealed by ChIP-seq here (Fig. 5e, Additional file

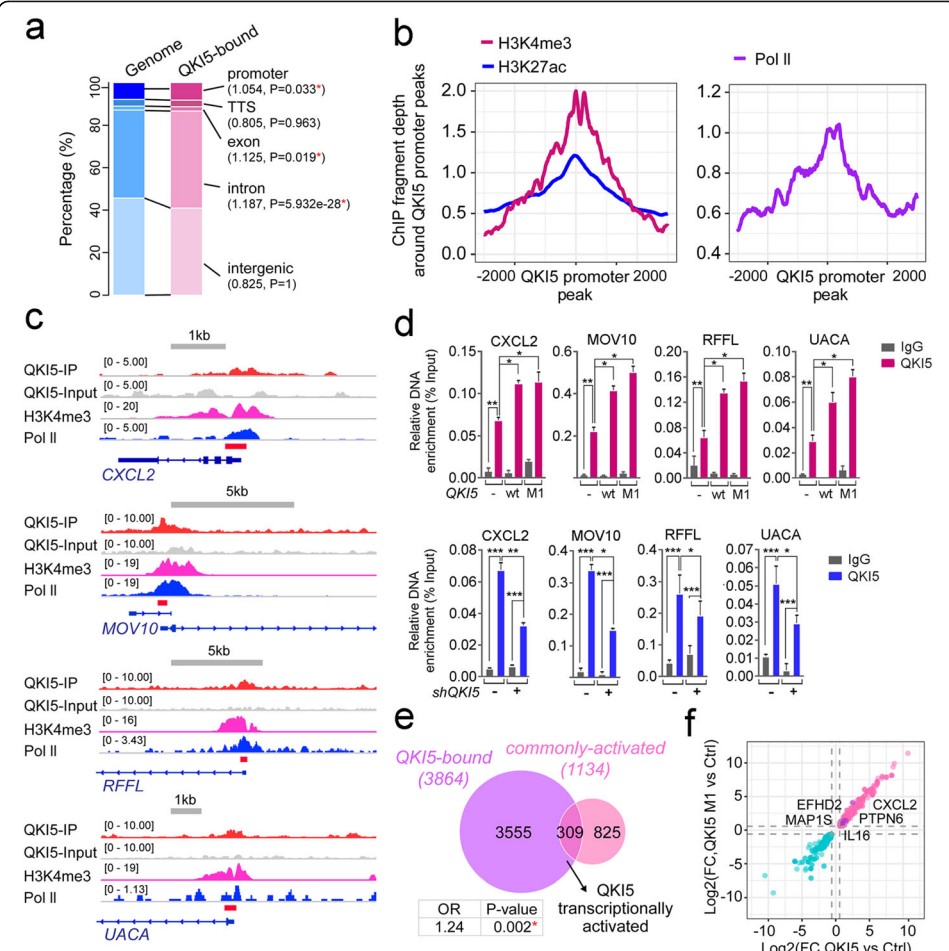

**Fig. 5** QKI5 preferentially binds to activated gene promoters. **a** Percentages of different genic and intergenic regions associated with QKI5 peaks as identified by ChIP-seq in THP-1 cells (QKI5-bound). The abundance of each type of region in the human genome (Genome) is shown for comparison. The numbers in the parentheses indicate the enrichment ratio relative to the genome; *P* values were evaluated by single-tailed Fisher's exact test (**P* value < 0.05). **b** Metaplot showing the distribution of H3K4me3 and H3K27ac ChIP-seq (left panel) and Pol II ChIP-seq fragment depth (right panel) within − 3000 bp to 3000 bp centered around QKI5 ChIP-seq peaks on promoter regions. **c** Genomic visualization of QKI5, H3K4me3, and Pol II ChIP-seq datasets on the indicated gene loci. **d** QKI5 ChIP-seq signal validation by ChIP-qPCR on the target gene loci in Ctrl- or QKI5/QKI5 M1-overexpressing cells (upper panel), as well as in shCtrl- or shQKI5-treated THP-1 cells (lower panel). Shown is the mean DNA enrichment relative to input; error bars indicate standard deviations around means from three biological replicates. Asterisks indicate significant differences between the specified samples (**P* value < 0.05, ***P* value < 0.01, *** *P* value < 0.001, *t* test). **e** Venn diagram illustrating the intersection of QKI5-bound genes and QKI5/QKI5 M1/PMA commonly activated genes, *P* value was evaluated by double-tailed Fisher's exact test (**P* value < 0.05). **f** Scatterplot shows log2 (fold change) of expression level of overlapping DEGs obtained from QKI5 and QKI5 M1-overexpressing cells, then filtered by the presence of QKI5 ChIP peaks. The purple dots indicate selected target genes

1: Figure S5b). We uncovered a significant overlap consistent with the proposed transcriptional regulatory role of QKI5 in our in vitro monocytic differentiation system (Fig. 5e, Additional file 1: Figure S5b and Additional file 7: Table S6). Indeed, GO enrichment analysis of the intersected genes generated from QKI5-bound and QKI5-regulated genes (commonly activated and commonly repressed genes) showed a distinct gathering in myeloid leukocyte-related pathways (Additional file 1: Figure S5c, Additional file 7: Table S6) [61–65].

Within the overlapping gene set, we found 309 candidate genes that were transcriptionally activated by QKI5 (Fig. 5e, Additional file 7: Table S6), of which several possessing essential roles during monocytic differentiation, including *CXCL2, microtubule associated protein 1S* (*MAP1S*), *protein tyrosine phosphatase non-receptor type 6* (*PTPN6*), *IL16*, and *EF-hand domain family member D2* (*EFHD2*) [66] (Fig. 5f). Taken together, we showed that QKI5 tended to enrich on genomic regions to regulate expression of target genes, including some that were involved in monocytic differentiation.

## Chromatin-associated QKI5 promotes target gene transcription

As a classic RBP, QKI5 might be recruited to the chromatin by its associating RNAs. To investigate whether the association of QKI5 to target loci was mediated by RNA, we performed an RNase-treated ChIP-qPCR analysis, which involved RNase-mediated RNA digestion prior to formaldehyde crosslinking, followed by routine ChIP-qPCR (Fig. 6a, Additional file 1: Figure S6a). Interestingly, QKI5's enrichment on target gene sites was not affected by RNA removal (Fig. 6b), indicating an RNA-irrelevant chromatin interaction. It was possible that QKI5 could also be recruited by other chromatin-interacting proteins, most likely TFs. To address this, we first screened for TF-binding motifs nearby QKI5's binding motif (Additional file 1: Figure S6b) using SpaMo [67] (see "Methods" for detail) and identified 96 distinct DNA elements within 150 bp of the QKI5 ChIP motifs (TGGGAYTA). Among them, 24 were annotated as hematopoiesis-related TF binding sites, including for the TF AP-1 (JUN), DNA-binding protein Ikaros (IKZF1), TF jun-B (JUNB), hypoxia-inducible factor 1-alpha (HIF1A), and thyroid hormone receptor alpha (THA) (Fig. 6c, Additional file 1: Figure S6c and Additional file 8: Table S7). Thus, it seemed likely that QKI5 might be brought to specific genomic loci by TFs.

To further verify the hypothesis of QKI5 recruitment by TFs, we performed co-IP with MS analysis to identify QKI5-interacting proteins (Fig. 6d, Additional file 1: Figure S6d, e and Additional file 8: Table S7). Gene ontology analysis showed that numerous DNA- and RNA-related proteins were identified, including 80 that were associated with the DNA related term (Fig. 6d, Additional file 8: Table S7). However, when we compared QKI5 motif neighbor TFs with all the QKI5-bound chromatin-interacting proteins, we did not find any shared candidates (Additional file 1: Figure S6f), suggesting that QKI5 might directly associate with chromatin or may be recruited to chromatin by other proteins/cofactors not identified by these assays, in addition to TFs.

Therefore, to determine whether QKI5 could bind DNA directly, we conducted DNA EMSA assay using biotin-labeled probes containing the QKI5 binding motif from the *CXCL2* promoter and QKI5 recombinant proteins. Results showed that QKI5 could interact with the wild-type *CXCL2* promoter as a shifting band was detected, but that this interaction was abolished if a mutant probe was used instead (Fig. 6e). Thus, QKI5 could recognize and bind directly to the specific DNA sequence from the *CXCL2* promoter in vitro.

We next detected QKI5's transcriptional regulatory activity in nuclei from QKI5-knockdown THP-1 cells (Additional file 1: Figure S6g) by employing the nuclear run-on (NRO) assay, which captured in situ newly transcribed products prior to their

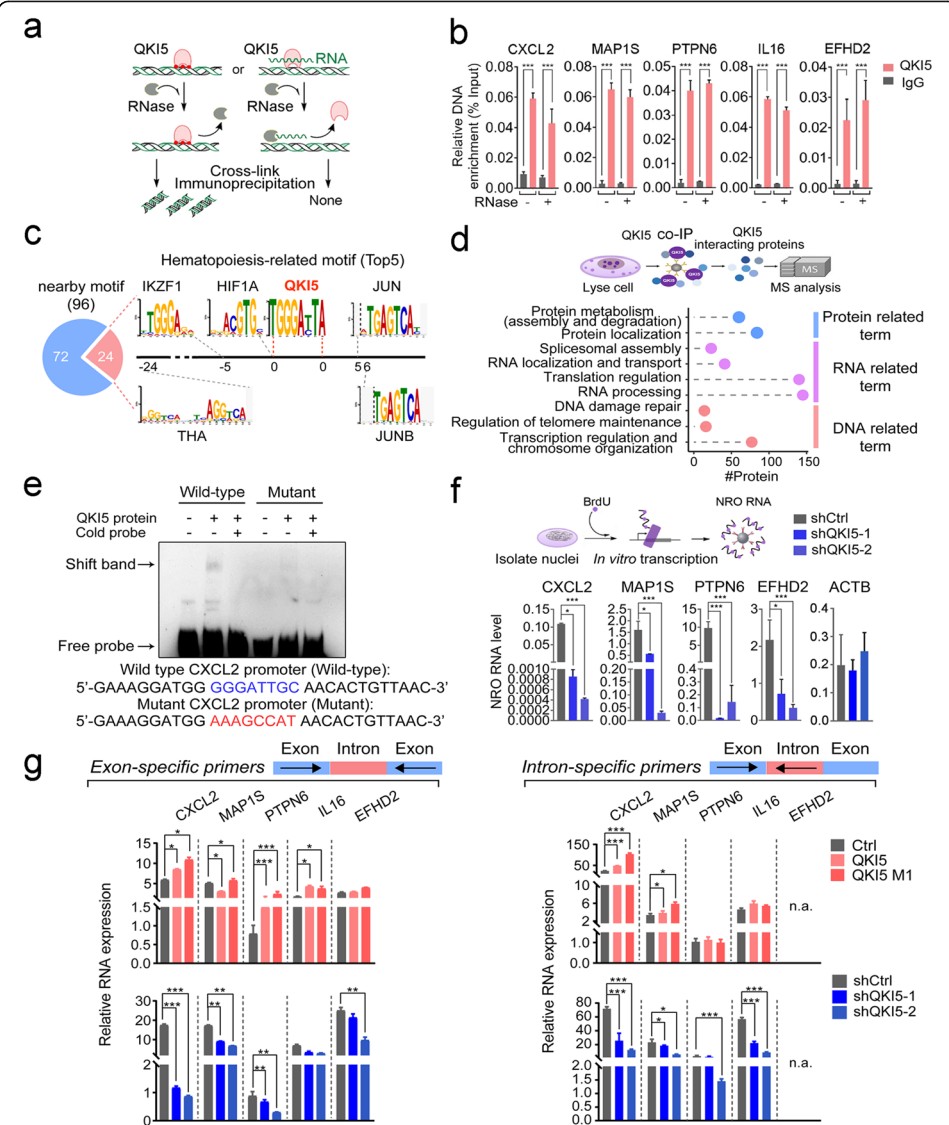

**Fig. 6** QKI5 binds to DNA and regulates target gene transcription. **a** Schematic diagram of RNase-ChIP assay that was applied to THP-1 cells. **b** ChIP-qPCR validation of QKI5 enrichment conducted on selected target genes with/without RNase treatment. Shown is the mean DNA enrichment relative to input; error bars indicate standard deviations around the means of three biological replicates. Asterisks indicate significant difference between indicated samples (***P value < 0.001, t test). **c** Nearby TF-binding motif prediction by SpaMo, which was used to identify putative hematopoiesis-related interaction partner TFs for QKI5. Top 5 of the identified TFs determined by e-value are shown in the figure. The number under axis represents best gap between QKI5 motif and indicated TF motifs. **d** Upper panel: Schematic diagram of QKI5's interaction partner screening. The interacting proteins were identified by QKI5 co-immunoprecipitation combined with mass spectrometry analysis. Lower panel: GO functional enrichment analysis of identified QKI5 interacting proteins. **e** In vitro association of QKI5 with the *CXCL2* promoter sequence identified by the DNA EMSA assay in which a 5′-biotin-labeled wild-type and mutant *CXCL2* promoter probes were used. The corresponding unlabeled ("cold") probes were used in the competitive assay. **f** Nuclear run-on assay in QKI5-knockdown THP-1 cells. Upper panel: Diagram of nuclear run-on (NRO) assay concept. Lower panel: NRO-qPCR validation of expression of selected target genes in shCtrl- or shQKI5-treated THP-1 cells. *ACTB* is a non-QKI5 target negative control. Error bars indicate standard deviations around the means of three biological replicates. Asterisks indicate a significant difference between the specified samples (*P value < 0.05, ***P value < 0.001, ns represents non-significant, t test). **g** RT-qPCR validation of the expression of selected target genes using exon-specific (left panel) and intron-specific (right panel) primers. qPCR was performed in Ctrl or QKI5/QKI5 M1-overexpressing (upper panel) and shCtrl- or shQKI5-treated (lower panel) THP-1 cells following by PMA induction for 48 h. Error bars indicate standard deviations around the means of three biological replicates. Asterisks indicate significant differences between the specified samples (*P value < 0.05, **P value < 0.01, ***P value < 0.001, n.a. represents not available, t test)

identification by RT-qPCR (Fig. 6f, upper panel). This clearly showed that reduced QKI5 level led to significant decreases of transcription activity at *CXCL2*, *EFHD2*, *MAP1S*, and *PTPN6* loci, while the transcription of non-QKI5-target gene like *actin beta* (*ACTB*) was comparable across all conditions (Fig. 6f, lower panel).

To further examine the effects of QKI5 on target genes' expression at the transcriptional level, we measured the abundance of both mature and nascent target transcripts using exon- and intron-specific primers (Fig. 6 g). Most of the target genes exhibited a significant increase in both exonic and intronic amplicons (except *EFHD2*) in either wild-type or mutant QKI5-overexpressing THP-1 cells at 0 h (Additional file 1: Figure S6h, upper panel) or 48 h (Fig. 6g, upper panel) of PMA induction. In contrast, QKI5 knockdown reduced the levels of these amplicons (Fig. 6g, lower panel; Additional file 1: Figure S6h, lower panel). That QKI5 could influence the level of nascent transcripts of target genes provided further evidence for the transcriptional regulatory potential of QKI5 in THP-1 cells.

Taken together, these results indicated the transcriptional regulatory activity of QKI5 in monocytic cells.

### QKI5 facilitates monocytic differentiation through activating *CXCL2* transcription

Given the central role of CXCL2 in monocytic differentiation [12–14], we further investigated the regulation of its expression by QKI5, and the functional relevance of this specific interaction. As revealed by dual-luciferase reporter assay (Fig. 7a-c), overexpression of either wild-type or mutant QKI5 (Additional file 1: Figure S7a) in 293T cells (used here to ensure high rates of transfection) increased the transcriptional activity of the *CXCL2* promoter (Fig. 7b, upper panel) whereas a decrease was observed in QKI5 knockdown cells (Fig. 7c, Additional file 1: Figure S7b). Meanwhile, the transcriptional activation effects brought about by QKI5 proteins were abolished in 293T cells bearing a mutant *CXCL2* promoter (Fig. 7b, lower panel). Additionally, levels of the CXCL2 protein also fluctuated consistently with QKI5 overexpression or knockdown in THP-1 cells (Fig. 7d, Additional file 1: Figure S7c).

To further understand the biological function of the QKI5/CXCL2 axis during monocytic differentiation, we performed "rescue" assays by employing either *CXCL2* shRNAs (shCXCL2) to diminish CXCL2 level ahead of QKI5 overexpression, or QKI5 shRNAs to knockdown QKI5 prior to CXCL2 overexpression (Fig. 7e). As expected, when CXCL2 was reduced (Additional file 1: Figure S7d), the differentiation-promoting ability of QKI5 was significantly decreased (Fig. 7f), suggesting the requirement of CXCL2 for QKI5's function. Next, the re-introduction of CXCL2 (Additional file 1: Figure S7e) partly restored the impaired monocytic differentiation caused by QKI5's decrease (Fig. 7g), further verifying that QKI5 promoted monocytic differentiation by regulating CXCL2 expression.

All these results allowed us to illustrate a novel regulatory pathway wherein QKI5 promoted monocytic differentiation via interacting directly with chromatin and activating the transcription of downstream functional genes (Fig. 7 h).

### Discussion

Proteins that bind RNA have typically been considered as functionally distinct from those that associate with DNA, and therefore have been studied separately. Thanks to

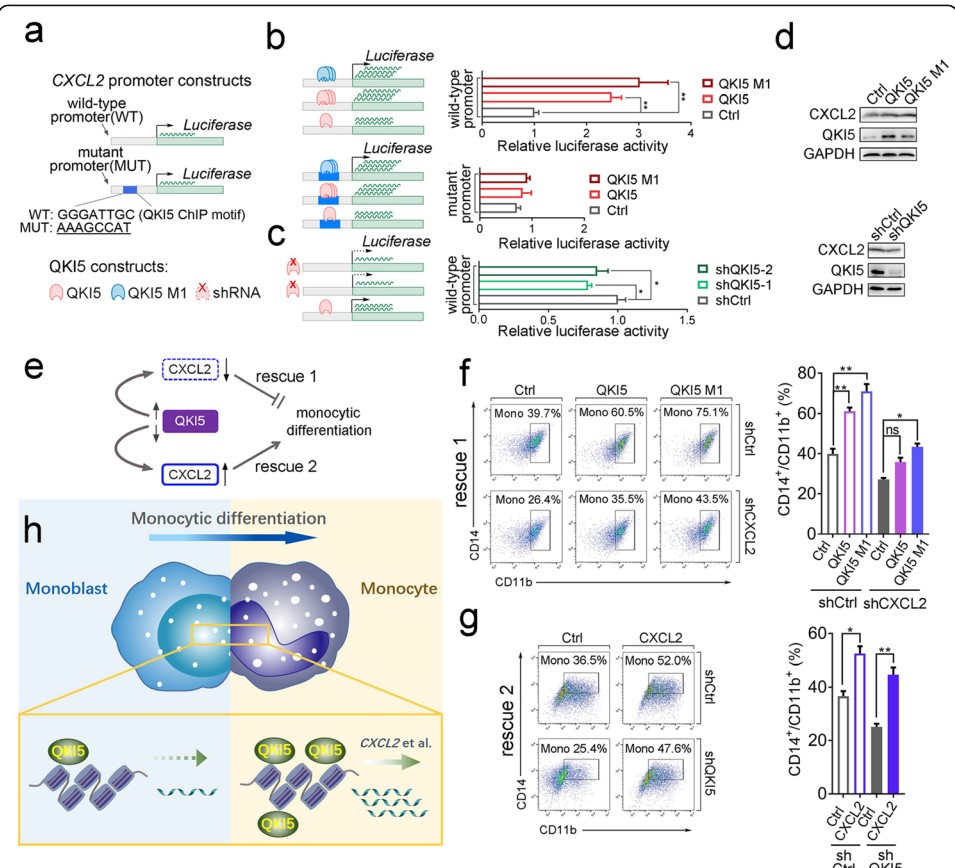

**Fig. 7** CXCL2 is a regulatory target of QKI5 during monocytic differentiation. **a–c** Dual-uciferase reporter assays. Three independent experiments were performed, and data are presented as mean relative luciferase activity ± SD. **a** Diagram of QKI5 constructs and *CXCL2* promoter constructs used. **b** The influence of QKI5/ QKI5 M1 overexpression on activity of the wild-type (upper panel) and mutant *CXCL2* promoters (lower panel). Left: Diagram of experiment. Right: 293T cells were co-transfected with the indicated combination of QKI5 and *CXCL2* promoter constructs or control construct, then the activity of the promoter was calculated by the ratio of *firefly* and *Renilla* luciferase activities. Error bars indicate standard deviations of the three biological replicates. Asterisks indicate significant differences between the specified samples (*$P$ value < 0.05, **$P$ value < 0.01, $t$ test). **c** The influence of QKI5 knockdown on activity of the wild-type *CXCL2* promoter. Left panel: Diagram of experiment. Right panel: The same assay as above was conducted with the combination of shCtrl or QKI5 shRNA and *CXCL2* promoter constructs or control construct. Asterisks indicate significant differences between the specified samples (*$P$ value < 0.05). **d** Immuno-blot of CXCL2 and QKI5 in Ctrl- or QKI5/QKI5 M1-overexpressing (upper panel) and shCtrl- or shQKI5-treated (lower panel) THP-1 cells. **e** Experimental design of the two sets of rescue assays. **f** Percentage of CD14$^+$/CD11b$^+$ cells among THP-1 cells within shCtrl- or shCXCL2-treated populations, detected by flow cytometry in rescue 1, in which THP-1 cells were treated with shCtrl- or shCXCL2- shRNAs, followed by Ctrl or QKI5 /QKI5 M1 transduction. Average percentage of CD14$^+$/CD11b$^+$ cells is shown on the right. Error bars indicate standard deviations around the means of three biological replicates. Asterisks indicate significant differences between the specified samples (*$P$ value < 0.05, **$P$ value < 0.01, ns represents non-significant, $t$ test). **g** Percentage of CD14$^+$/CD11b$^+$ cells among THP-1 cells within shCtrl- or shQKI5- treated population detected by flow cytometry in rescue 2, in which THP-1 cells were treated with shCtrl- or shQKI5- shRNAs, followed by Ctrl or CXCL2 transduction. Average percentage of CD14$^+$/CD11b$^+$ cells is shown on the right. **h** Schematic diagram of QKI5's transcriptional regulatory function during monocytic differentiation

rapid advances in technology, numerous RNA-chromatin interactions have recently been revealed [68, 69], which has led to the discovery of several DNA-associated RBPs [3–8]. What remains unclear is just how many RBPs also bind DNA, and whether their role is restricted to mediating lncRNAs' functions or rather extends to independent activity at the chromatin interface. Moreover, the underrated dual RNA-/DNA-binding

capacity that has been associated with a growing number of proteins has now forced us to adopt a more holistic view of chromatin-associating RBPs and their possible regulatory functions.

Here, we conducted a comprehensive screening to identify all Che-RBPs in two human hematopoietic cell lines (THP-1 and K562), and in 293T cells. Interestingly, over half of the Che-RBPs we characterized were bound to chromatin in all three cell lines tested, suggesting that Che-RBPs may be intrinsic components of the chromatin environment and play fundamental regulatory roles in gene expression. Moreover, analyzing the ChIP-seq, CLIP-seq, and RNA-seq data sets of 7 hematopoiesis-related Che-RBPs (hChe-RBPs) enabled us to compare their DNA- and RNA-binding features with their effects on genes' expression.

We found that the overlapping rates between ChIP-seq and CLIP-seq signals were very low of each hChe-RBPs. The most likely case is that hChe-RBPs interact with chromatin or RNA via different manners. As for the presumable RNA-mediated chromatin interactions, our results show that the interaction of hChe-RBPs to chromatin is neither mediated by newly transcribed RNAs, nor by neighbor nascent transcripts because the distances between these RBPs' DNA and RNA binding sites are mostly more than 5 kb (Fig. 2f). Therefore, their interactions might be mediated by ncRNAs like LncRNAs which could function at long distance, or by the high-ordered structures of chromatin.

Moreover, when comparing the DNA/RNA interaction patterns demonstrated by integrated analyses of ChIP-seq/CLIP-seq datasets with RNase-treated subcellular fractionation results, we can conclude that the hChe-RBPs presenting DNA/RNA separated or combined chromatin interaction mode accord with their different DNA- or RNA-binding tendency. The separated chromatin and RNA interaction of hChe-RBPs indicates that they might possess unique and unrelated regulatory function at DNA or RNA level.

According to our results, besides RBPs' classical regulations on RNA, the hChe-RBPs could also locate on chromatin and possess transcriptional regulatory potential. Among them, QKI5, KHSRP, and SETD1A tended to regulate genes' expression by binding to genomic regions of target genes (Fig. 2j). Notably, SETD1A is also a known methyltransferase possessing DNA-binding domain [70]; KHSRP has been reported to have DNA-binding domain [71]; as for QKI5, although there's no reports on known DNA-binding domains on QKI5, recent studies have shown that it could be recruited to chromatin carrying out downstream regulatory functions by transcription factors [72, 73]. Besides, other hChe-RBPs tested in our study have not been reported to contain any known DNA-binding domains. As above, hChe-RBPs could be multifunctioning factors regulating genes' expression at different levels. And for transcriptional regulation, they might function by direct (with DNA-binding motif) or indirect (without known DNA-binding domain, coordinate with other factors) interactions with chromatin.

The large number of Che-RBPs that we identified has extended this property from the small set of well-studied RBPs to a more general group that is now ripe for future mechanistic study. This subclass of RBPs might be considered as functionally superior to their pure RNA-binding cousins, as they may fine-tune genes' expression via association with DNA or with RNA products, acting at both transcriptional and post-transcriptional levels. In addition, we might also speculate that competitive RNA- and

DNA-binding activity by some Che-RBPs could provide an additional regulatory layer beyond transcriptional and post-transcriptional levels. This study focused predominantly on hematopoietic Che-RBPs, but the same approach could be readily adapted to screen for novel Che-RBPs in other cell types and systems. As we now know that some RBPs locate on chromatin co-transcriptionally [74], there should also be further investigation into the extent of Che-RBPs' regulation of chromatin through genetic perturbations and functional experiments on individual Che-RBP to test their effects on proximal gene transcription and RNA transcript processing.

One of the most promising candidate hChe-RBPs identified by our initial screening is QKI5, which is known to bind RNA and regulate RNA splicing at the post-transcriptional level in mouse oligodendrocytes and neonatal brain as well as in the human erythropoietic cell lines K562 and HEL [7, 8, 52, 75]. Here we demonstrate that QKI5 is a potent Che-RBP that regulates genes' transcription at specific loci in hematopoietic monocytic cells. While some RBPs, such as AGO1 and METTL3, are recruited to chromatin via RNA or proteins [4, 76], others, such as Lin28A, bind chromatin through a Cold Shock Domain, which is able to interact with both single-stranded RNA and DNA [3]. Transcriptome analysis combined with experimental validation of selected target genes have showed that QKI5 could act as a transcription activator independent of RNA or TFs we screened, yet the question of whether QKI5 is assisted by other factors beyond our screening still remains open. Since QKI5 does not contain any known DNA-binding domains, future studies might also uncover some novel DNA-binding structures within this fascinating protein.

Having defined the molecular features of QKI5's interaction with DNA, we want to understand the potential significance of this interaction in hematopoietic differentiation. De Bruin et al. previously reported that QKI5 protein was elevated during monocytic to macrophage differentiation in human peripheral blood (PB), and also in THP-1 cells differentiating in vitro [10], while Fu et al.'s work showed a delayed effect of QKI5 during monocytic-macrophage differentiation in both progenitor cells from human cord blood and HL-60 cells [11]. While these data provides further support to the functional role of QKI5 in human hematopoietic cells, as both studies have focused purely on QKI5's RNA-binding-related capacity, it is yet unclear whether the observed effects were mediated at DNA or RNA transcripts level. Our work here suggests that transcriptional regulation of QKI5 plays a dominant role in this system. Overall, we speculate that Che-RBPs, such as QKI5, are likely to work at both the transcriptional and post-transcriptional levels, orchestrating gene expression whilst fine tuning its effects prior to final protein expression.

Taken together, here we characterize a new subpopulation of RBPs named Che-RBPs, which are enriched on chromatin and have functional roles in human cells. Seemingly, Che-RBPs along with RNAs and other DNA-binding proteins may synergize to form integrated regulatory machines that fine-tune genes' expression. Furthermore, we could speculate that these regulatory machines may also help to establish active/repressive transcriptional regions or domains in the nucleus for the construction of higher-order chromatin structures.

## Conclusions

In this study, we first obtain the full view of RBPs' distribution pattern in nucleus and identify a set of RBPs enriched on the chromatin (Che-RBPs) by using a convenient

screening strategy. Combining ChIP-seq, CLIP-seq, and RNA-seq analysis of hematopoietic Che-RBPs (hChe-RBPs), we characterize the transcriptional regulatory potential of such hChe-RBPs. Moreover, we also identify an unexpected novel transcriptional regulatory role of RBP QKI5, which deposits on genetic regions of functional target genes to activate their transcription thus promoting monocytic differentiation.

## Methods

### Cell culture

THP-1 was purchased from the cell resource center of Shanghai Institutes for Biological Science, and 293T and K562 were purchased from the cell resource center of Institutes of Basic Medical Sciences, Chinese Academy of Medical Sciences. THP-1 and K562 were cultured in PRMI 1640 medium (Gibco, Carlsbad, CA, USA); 293T cells were cultured in Dulbecco's modified Eagle's medium (DMEM) (Gibco). All cultures were supplemented with 10% fetal bovine serum (FBS) (Gibco) and 100 U/ml Penicillin Streptomycin mixtures (10000 U/ml, Gibco) at 37 °C in 5% $CO_2$. The monocytic differentiation of THP-1 was induced with PMA (Sigma-Aldrich, Deisenhofen, Germany) at a final concentration of 10 nM. Human umbilical cord blood (UCB) was obtained from normal full-term deliveries from Haidian Maternal & Child Health Hospital (Beijing, China). $CD34^+$ cells were enriched from mononuclear cells (MNCs) through positive immunomagnetic selection (CD34 MultiSort kit, Miltenyi Biotec, Bergisch-Glad-bach, Germany). The isolated $CD34^+$ hematopoietic stem/progenitor cells (HSPCs) were cultured in IMDM supplemented with 20% serum substitute (BIT; Stem Cell Technologies, Vancouver, BC, Canada), 100 μM 2-ME, 2 ng/ml recombinant human IL-3, 100 ng/ml recombinant human SCF (Stem Cell Technologies), 50 ng/ml recombinant human M-CSF (PeproTech, Rocky Hill, NJ, USA), 10 ng/ml recombinant human IL-6 (PeproTech), 100 ng/ml recombinant human Flt-3 (PeproTech), 100 U/ml Penicillin Streptomycin mixtures (10,000 U/ml, Gibco), and 2 mM L-glutamine (200 mM, Gibco). Cells were harvested every 3–5 days.

### RNA extraction, reverse transcription, and quantitative real-time PCR

Total RNA was extracted from cell samples using Trizol reagent (Invitrogen, Carlsbad, CA, USA) according to the manufacturer's instructions. Approximately 1–4 μg of total RNA was used to generate cDNA by M-MLV reverse transcriptase (Invitrogen). Oligo (dT)18 or random primer (Promega, Madison, WI, USA) was used for reverse transcription of mRNAs. Quantitative real-time PCR (qPCR) was carried out using Bio-Rad CFX-96 (Bio-Rad, Foster City, CA, USA) in triplicates. The data were normalized to GAPDH mRNA expression. All RT-qPCR primers used in this study are listed in Additional file 9: Table S8.

### Subcellular fractionation

$6 \times 10^6$ cells were washed in PBS and suspended in 400 μl Solution A (10 mM HEPES 7.9, 10 mM KCl, 1.5 mM MgCl2, 0.34 M sucrose, 10% glycerol, 1 mM DTT, 1 × protease cocktail (Roche Life Science, Indianapolis, IN, USA)). Two microliters 20% Triton X-100 were added to a final concentration of 0.1%, mixed gently and incubated on ice

for 5 min. The cytoplasmic (S1) and nuclear fractions were harvested by centrifugation at 1300×g for 4 min. The isolated nuclei were washed in 1 ml Solution A and lysed in 400 μl Solution B (3 mM EDTA, 0.2 mM EGTA, 1 mM DTT, and 1× protease cocktail (Roche Life Science) and incubated on ice for 30 min. The soluble nuclear extract (SNE) and chromatin-pellet extract (CPE) fractions were separated by centrifugation at 1700×g for 4 min. RNase-treated samples were subjected to the reported protocol [77] with some modifications: in brief, in the last step of fractionation, chromatin fractions were incubated with 20 μg/ml RNase A in PBS for 5 min at 4 °C, while RNase A-free samples were incubated with PBS alone.

### Protein mass spectrometry (MS)

The SNE and CPE fractions were prepared from THP-1 cells, K562 cells, and 293T cells as above. The lysates from each fraction were subjected to SDS/PAGE (10% separation gel) and dyed with 0.25% Coomassie Brilliant Blue (Thermo Fisher Scientific, Waltham, MA, USA). The stained protein bands were cut off from the gel and sent to Protein Research and Technology Center (Tsinghua University, Beijing, China) for mass spectrometry (LC MS/MS) analysis. Briefly, proteins were subjected to in-gel tryptic digestion followed by peptide desalting and concentration before mass spectrometry detection.

### RNA-seq

Total RNA was extracted from cell samples using Trizol reagent (Invitrogen, Carlsbad, CA, USA) as previously mentioned. The RNA was amplified and subjected to 150 bp paired-end deep sequencing on the Illumina HiSeq X Ten and NovaSeq 6000 platforms by Novagene (Beijing). The plasmids of shRNA targeting indicated proteins were purchased from Origene (Origene Technologies, Rockville, MD, USA).

### Plasmid construction

For QKI5 overexpression, the human QKI5 (NM_006775) cDNA ORF clone was purchased from Origene (Origene Technologies, Rockville, MD, USA) and sub-cloned into pCDH (QKI5) (System Biosciences, Palo Alto, CA, USA). QKI5$^{V157E}$ (QKI5 M1) on the KH domain of QKI5 cDNA was created using a QuickChange Site-Directed Mutagenesis kit (Agilent Technologies, La Jolla, CA, USA). The short hairpin RNA (shRNA) plasmids targeting indicated proteins were purchased from Origene (Origene Technologies, Rockville, MD, USA). All the primers and the information of shRNA plasmids used in this study are listed in Additional file 9: Table S8.

### Lentivirus production and cell infection

Recombination lentiviruses for QKI5 overexpression were produced using pCDH-based constructs and lentiviruses for knockdown of indicated proteins were produced using shRNA plasmids purchased from Origene (Origene Technologies, Rockville, MD, USA). Matching lentivirus packaging was purchased from System Biosciences (System Biosciences, Palo Alto, CA, USA) and used according to the manufacturer's instructions. Harvested viral particles were added to HSPCs or THP-1 cells in 6-well plates containing 8 μg/ml polybrene (Sigma-Aldrich, Deisenhofen, Germany). After 12 h of

incubation, the cells were refreshed with complete medium and subjected to the following experiments.

### Flow cytometry analysis

The HSPCs or THP-1 cells were induced towards monocytic differentiation and harvested at different time points. The cells were rinsed twice with PBS and resuspended in 100 μl PBS. Then, the cells were incubated with PE-conjugated anti-CD14 and APC-conjugated anti-CD11b from eBioscience (Thermo Fisher Scientific, Waltham, MA, USA) at 4 °C for 30 min. After the incubation, cells were washed with 1 ml PBS, resuspended in 100 μl 4% PFA (Solarbio, Beijing, China), and analyzed immediately using an AccuriC6 flow cytometer (BD Biosciences, San Jose, CA, USA).

### Colony-forming unit assay

Lentivirus-infected HSPCs were plated in 35-mm petri dishes containing 2 mL methyl-cellulose medium (Stem Cell Technologies, Vancouver, BC, Canada) containing rh SCF, rh GM-CSF, rh IL-3, rh IL-6, and rh G-CSF. The cells were incubated at 37 °C with 5% $CO_2$ for 10 days for CFU-M quantification.

### Chromatin immunoprecipitation (ChIP)

$1 \times 10^7$ THP-1 cells per sample were crosslinked with 1% formaldehyde for 15 min. Crosslinking was neutralized with 0.125 M glycine, and cells were rinsed in PBS twice. Then chromatin was sonicated using a Diagenode Bioruptor (Diagenode, Seraing, Belgium) for 30 min with 30 s pulse/pause cycles in polycarbonate tubes on ice to break chromatin into 200- to 500-bp fragments. Unbroken debris was spun down, and then the chromatin was split into two equal portions. One was used for control IgG antibody (Millipore, Darmstadt, Germany), and the other portion was incubated with indicated antibody listed in Additional file 9: Table S8. Salmon sperm-coated protein A/G beads (Millipore) were added to the two portions of the chromatin with equal volume. Then, the mixture of chromatin-antibody-protein-A/G beads was incubated overnight at 4 °C. After washing 4 times, immunoprecipitated DNA was eluted from beads and purified for subsequent qPCR test. For the ChIP assay with RNase A treatment, the experiment was conducted as described previously with minor modifications [78]. Briefly, $1 \times 10^7$ THP-1 cells were collected by centrifugation, permeabilized in 1.5 ml PBT (PBS; 0.05% Tween 20), and treated with 80 μg/ml RNase A (Thermo Fisher Scientific, Waltham, MA, USA), or 100 U/ml RNase inhibitor (Promega, Madison, WI, USA) for 45 min at 37 °C followed by crosslinking with 1% formaldehyde for 10 min at RT. Crosslinked chromatin was isolated, fragmented, and immunoprecipitated as mentioned above. Immunoprecipitated DNA was purified and analyzed by qPCR. All the ChIP-qPCR primers used in this study are listed in Additional file 9: Table S8. For ChIP-seq, DNA fragments immunoprecipitated from ChIP assay were purified for subsequent library preparation using The NEBNext Ultra II DNA Library Prep (New England Biolabs, Ipswich, MA, USA) according to the manufacturer's instructions. The DNA fragments were amplified and subjected to 150 bp paired-end deep sequencing on the Illumina HiSeq X Ten and NovaSeq 6000 platforms by Novagene (Beijing).

### Enhanced UV crosslinking, immunoprecipitation, and high-throughput sequencing (eCLIP-seq)

eCLIP was performed as described previously with minor modifications [79]. Briefly, cells were UV-crosslinked at 150 mJ and 254 nm wavelength in 10-cm plates with 3 ml of cold PBS. Then cells were pelleted, flash frozen in liquid nitrogen, and stored at − 80 °C. The pellet was lysed with lysis buffer (50 mMTris-HCl, pH 7.4; 100 mM NaCl; 1% NP-40; 0.1% SDS; 0.5% sodium deoxycholate; 1X protease inhibitor cocktail, Roche) followed by further RNase I, and Turbo DNase treatment as described. The lysate was incubated with specific antibody overnight at 4 °C for immunoprecipitation. Forty microliters of protein A beads (Invitrogen, USA) was added and incubated for 2 h followed by washes as described. Following end repair and 3′ adaptor ligation, size selection was conducted using Nupage 4–12% Bis-Tris protein gels followed by transfer to nitrocellulose membranes. RNAs on nitrocellulose were harvested and reverse transcribed using SuperScript III (Thermo Fisher, USA). cDNA libraries were then prepared as described and sequenced by using Illumina HiSeq X Ten and NovaSeq 6000 with strand-specific paired-end 150 bp read length.

### Nuclear Run-On assay

A nuclear run-on assay was performed as described [80]. Approximately 1 million nuclei were resuspended in nuclei storage buffer (50 mM Tris-HCl, pH 7.8, 5 mM $MgCl_2$, 20% glycerol, 1 mM DTT, 0.44 M sucrose) and then mixed with an equal volume of reaction buffer (50 mM Tris-HCl, pH 7.5, 5 mM $MgCl_2$, 1 mM DTT, 150 mM KCl, 0.2% sarkosyl (Sigma-Aldrich, Deisenhofen, Germany), 40 units RNase inhibitor, 1 mM ATP, 1 mM GTP, 1 mM CTP, and 0.5 mM 5-BrUTP (Sigma-Aldrich)). After incubation at 30 °C for 10 min, the reaction was stopped by adding Trizol reagent (Invitrogen, Carlsbad, CA, USA). RNAs were extracted and treated with DNase I (Promega) to remove genomic DNA. The purified RNAs were incubated with 2 mg anti-BrdU antibody (Abcam, London, UK) at 4 °C for 2 h and then subjected to immunoprecipitation with Dynabeads Protein G (Invitrogen,) for 1 h. Precipitated RNAs were extracted by Trizol reagent and used for subsequent manipulations.

### DNA EMSA

The biotin-labeled wild-type and mutant *CXCL2* promoter probes, as well as corresponding cold probes, were synthesized by Thermo Fisher Scientific company (Thermo Fisher Scientific, Waltham, MA, USA). A total 50 fmol of biotin-labeled DNA probes were incubated with 4 μl QKI5 recombinant protein purchased from Origene (Origene Technologies, Rockville, MD, USA) using the LightShift Chemiluminescent EMSA Kit (Pierce, IL, USA) according to the manufacturer's protocol.

Competition experiments were performed with 200-fold molar excess of the unlabeled probes (cold probe) preincubation. The reactions were incubated at room temperature for 20 min before adding DNA loading dye and separated by native 8% PAGE. The probes used for the DNA EMSA experiment are listed in Additional file 9: Table S8.

### Immunoprecipitation

THP1 cells were collected and subcellular fractionation was used at first to extract nuclei. Then the nuclei were lysed with RIPA buffer and subjected to incubation with

specific antibody or IgG and protein A beads (Invitrogen, USA) overnight at 4 °C for immunoprecipitation. After 3 washes with RIPA buffer, 1× SDS loading buffer was added to each sample and boiled at 95 °C for 5 min to elute proteins from beads, before subjecting them to the following processes.

### Western blot

Cell lysates were subjected to SDS/PAGE (10% separation gel) and transferred onto a PVDF membrane. Primary antibodies against indicated proteins were used followed by incubation with horseradish peroxidase conjugated secondary antibodies. Signals were detected using an ECL (enhanced chemiluminescence) kit (Millipore, Darmstadt, Germany). The information of primary antibodies and secondary antibodies is listed in Additional file 9: Table S8.

### Luciferase reporter assay

293T cells were co-transfected with pRL-TK, pGL3-CXCL2 wild-type/mutant promoter constructs, and pCMV6-QKI5/ QKI5 M1 constructs or shQKI5 plasmids using Lipofectamine LTX (Invitrogen, Carlsbad, CA, USA) in a 24-well plate. The plasmid pRL-TK containing Renila luciferase was used as an internal control. The transfection medium was replaced with complete medium after 5–6 h and the cells were cultured at 37 °C in 5% $CO_2$ for an additional 24–48 h. The cells were harvested, and luciferase activity was measured using a dual-luciferase assay system (Promega, Madison, USA) according to the manufacturer's instructions. The shQKI5 plasmids were purchased from Origene (OriGene Technologies, Rockville, MD, USA).

### Rescue assay

For rescue 1, THP-1 cells were firstly transduced with shRNA plasmids of CXCL2. After 24 h, the cells were transduced with QKI5 /QKI5 M1-overexpressing plasmids or control along with PMA induction for another 48 h. For rescue 2, THP-1 cells were firstly transduced with shRNA plasmids of QKI5, and 24 h later, cells were transduced with CXCL2-overexpressing plasmid or control along with PMA induction for another 48 h. CD14 and CD11b expression in THP-1 cells undergoing monocytic differentiation was analyzed by flow cytometry. CXCL2 and QKI5 shRNA plasmids were purchased from Origene (OriGene Technologies, Rockville, MD, USA).

### Antibodies and reagents

All information on antibodies and reagents is listed in Additional file 9: Table S8.

### Bioinformatic analysis

#### MS data analysis

The mass spectrometry data was analyzed by Proteome Discoverer 1.4 (Thermo Fisher Scientific) supported by Protein Research and Technology Center (Tsinghua University, Beijing, China). The protein score was used to estimate the reliability of protein identification which was evaluated by PSMs (peptide spectrum matches—the number of identified peptides spectra matched for the proteins in second-order MS), matching

rate, and peptide FDR confidence level. The area value represented the relative quantity of protein defined by the peak area of peptides on chromatography.

For overall MS data analysis, we retained the protein included in the Swiss-Prot reviewed section of Uniport [81] database (https://www.uniprot.org/). The repeatability between two biological replicates was evaluated by Pearson correlation coefficient with area value. For MS data analysis in Fig. 1, we filtered out proteins with a score < 10. The area ratio was calculated by the ratio of average area value of proteins in the CPE and SNE fractions, and represented the relative enrichment of each protein in the CPE fraction. Combined with the protein score and the area ratio, proteins were divided into three classifications: CPE enriched proteins (area ratio ≥ 1.5), SNE enriched proteins (area ratio ≤ 0.67), and colocalized proteins (0.67 < area ratio < 1.5). For analysis of MS data generated from QKI5-co-immunoprecipitation assay in Fig. 6, we retained the QKI5-interacting proteins with a score > 0 and area ratio (IP/IgG) > 1.5.

The function of commonly CPE enriched proteins in three cell lines was manually classified based on molecule function annotation in DAVID database. Classification of protein domain was referred to RBPDB (http://rbpdb.ccbr.utoronto.ca/) [17] and DAVID database (https://david.ncifcrf.gov/) [82], and the gene functional network was generated by Cytoscape [83].

### ChIP-seq dataset analysis

The ChIP-seq datasets of histone modifications (H3K4me3, H3K27ac, H3K27me3, H3K79me2, H3K36me3) and Pol II were downloaded from the European Bioinformatics Institute (http://wwwdev.ebi.ac.uk/) [84]. Details of these data sources are listed in Additional file 10: Table S9.

For overall ChIP-seq datasets, reads were aligned to the Homo sapiens genome (Ensembl GRCh38.p5) using Bowtie2 [85] PE mode with default parameters. Only reads with mapping quality score > 30 were retained for further analysis by SAMtools [86]. Two biological replicates were merged to create the "Tag Directory" file by "makeTag-Directory." Peak finding and downstream data analysis were performed using HOMER software by "findPeaks" [87]. For hChe-RBP ChIP-seq datasets, we used the "factor" mode with the parameter "-tbp 1 -inputtbp 1 -F 2.5 -P 0.00001 -L 2.5 -LP 0.02 -ntag-Threshold 3.5." For histone modification ChIP-seq and Pol II ChIP-seq datasets, we used "histone" mode and "factor" mode with default parameter separately.

Enrichment of RBPs' DNA-binding sites on the genomic region was calculated by HOMER "annotatePeaks.pl." The repeatability between two biological replicates was evaluated by Pearson correlation coefficient with read coverages for genomic regions per 1000 bp, which generated from Homer "getPeakTags." The histone modifications and Pol II ChIP fragment depth around QKI5 promoter peaks (from – 3000 bp to 3000 bp) was made by HOMER "annotatePeaks.pl." Overlapped peaks number between each group of ChIP-seq datasets were determined by BedTools "intersectBed" [88]. The genomic visualization of hChe-RBPs ChIP-seq datasets at the indicated gene locus was generated by Integrative Genomics Viewer (IGV) software [89]. MEME-ChIP tools on MEME online suite (http://meme-suite.org/) was used for QKI5 DNA-binding motif discovery coupled with an e-value to determine motif enrichment and significance [90]. Nearby TF motifs to QKI5 motif (TGGGAYTA) was determined by SpaMo tools on

MEME online suite [67] with an e-value to determine motif enrichment and significance. Motif enrichments were both based on Hoocomoco11 (http://hocomoco11.autosome.ru/) [19] core database.

### CLIP-seq dataset analysis

The hChe-RBPs CLIP-seq datasets were processed in accordance with previous studies [79], and the eCLIP-seq data processing pipeline was available at "https://github.com/YeoLab/eclip." The Raw reads with distinct inline barcodes were demultiplexed using in-house scripts, and the 10-mer random sequence was appended to the reads name in bam file for later usage. Low-quality reads and adapter sequence were trimmed by cutadapt [91]. Repetitive reads were removed by aligning reads with human repetitive element sequence on RepBase database (https://www.girinst.org/) by STAR. Cleaned reads were mapped to Homo sapiens genome (Ensembl GRCh38.p5) by STAR [92]. PCR duplicate reads were removed by in-house script based on sharing identical random sequence. Two biological replicates were merged by SAMtools "merge" for following analysis. The CLIP-seq data processing statistics are listed in Additional file 10: Table S9. Peak calling and downstream data analysis were performed using clipper [93] software. Peak normalization was performed by "Peak_input_normalization_wrapper.pl" tools, available at "https://github.com/YeoLab/eclip." The CLIP-seq peaks were filtered by $P$ value $< 10e{-}3$ and fold change $> 4$.

Enrichment of hChe-RBP's RNA binding sites on the human genomic region was calculated by R package (ChIPseeker) [94]. The repeatability between two biological replicates was evaluated by Pearson correlation coefficient with read coverages for genomic regions per 1000 bp, which were generated from Deeptools "multiBamSummary" [95].

The gene type classification was based on the Vega website (http://vega.archive.ensembl.org/info/about/gene_and_transcript_types.html) and previous study (Additional file 2: Table S1) [7]. The relative distance between ChIP-seq peaks and neighboring CLIP-seq peaks was determined by BedTools "closest." MEME-ChIP tools on MEME online suite (http://meme-suite.org/) was used for hChe-RBPs RNA-binding motif discovery coupled with an e-value to determine motif enrichment and significance with standard RNA alphabet [90]. The list of house-keeping genes and cell-type-specific genes in THP-1 cells were referred to the Human Protein Atlas database (https://www.proteinatlas.org/). House-keeping genes are expressed in all 69 cell lines used in the Cell Atlas. The THP-1 cell-specific genes were included in the enriched genes, group enriched genes, and enhanced genes categories of the THP-1 cell line.

### RNA-seq dataset analysis

RNA-seq reads were aligned to Homo sapiens genome (Ensembl GRCh38.p5) using Tophat2 [96] PE mode with the default parameters, and uniquely mapped reads were retained for further analysis filtered by SAMtools. HTSeq-counts [97] were used for calculating gene counts, and then normalized for transcript per million (TPM) using in-house scripts. Differential gene expression analysis was conducted using DESeq2 [98] with the $P$ value $< 0.05$ and $|\log2(\text{fold change})| > \log2 (1.5)$ in Fig. 4, and $P$ value $< 0.05$ and $|\log2(\text{fold change})| > \log2 (1.2)$ in Fig. 2.

### Gene ontology functional enrichment analysis

Gene ontology functional enrichment analysis was performed by R package (clusterProfiler) [99], and we mainly focused on "biological process." Terms with $P$ value $< 0.05$ were considered to be enriched.

### Reference datasets

The RBP dataset was supported by RBPDB (http://rbpdb.ccbr.utoronto.ca/) [17] and ATtRACT (http://attract.cnic.es) [16]. The TF dataset was collected in JASPAR (http://jaspar.genereg.net/) [18] and TFs with ChIP-seq datasets supported in Hoocomoco11 (http://hocomoco11.autosome.ru/) [19] full database. Furthermore, the gene names and Ensembl IDs in these databases have been corrected manually. The hematopoiesis-related proteins were collected from the Gene Ontology database (http://geneontology.org/) [22], with the following keywords: hemopoiesis; hematopoietic; erythrocyte; megakaryocyte; monocyte; neutrophils; lymphocyte; myeloid. This term was also supplemented by nearly 10 years of research on PubMed [9, 23–48] related to hematopoietic processes.

## Statistical analysis

Data were presented as the mean ± SD with the error bar. For qPCR, dual-luciferase assays, and FACS results, statistical analyses were performed using double-tailed Student's $t$ test at a confidence interval of 95%.

In Fig. 2j, Fig. 5e, Additional file 1: Figure S5b and Figure S6f, double-tailed Fisher's exact tests were performed to calculate $P$ values at a confidence interval of 95%. In Fig. 5a, single-tailed Fisher's exact tests were performed to calculate $P$ values. In Fig. 4c, d, Additional file 1: Figure S1b and Figure S2c and Figure S6e, Pearson correlation coefficient was used to estimate the repeatability between the samples or replicates. Jaccard index was used to assess similarity of occupied genes between ChIP-seq and CLIP-seq datasets. Whenever asterisks are used to indicate statistical significance, *stands for $P < 0.05$; **$P < 0.01$, ***$P < 0.001$, and ****$P < 0.0001$. The n.a. represents "not available," and ns represents "non-significant." All statistical analyses were done in R and Prism.

## Supplementary Information

---

**Additional file 1: Figure S1-S7** and raw blot images in Figures and Supplementary Figures

**Additional file 2: Table S1** RBP&TFs library and hematopoiesis-related RBPs

**Additional file 3: Table S2** Sub-nuclear distribution of RBPs and TFs in 3 cell lines (THP-1, K562, 293T)

**Additional file 4: Table S3** Characteristics of common CPE enriched protein in 3 cell lines (THP-1, K562, 293T)

**Additional file 5: Table S4** RBP transcriptionally regulated genes and GO functional annotation

**Additional file 6: Table S5** DEGs results and GO term of PMA, QKI5, QKI5 M1 RNA-seq

**Additional file 7: Table S6** QKI5 transcriptionally regulated genes

**Additional file 8: Table S7** QKI5 motif nearby TF motif and QKI5 co-IP mass spectrometry (MS) data

**Additional file 9: Table S8** Primers and oligos, antibodies and reagents used in this study

**Additional file 10: Table S9** Sequencing datasets

**Additional file 11.** Review history

---

### Acknowledgements

We thank Insight Editing London for language polishing and editing of the revised manuscript.

**Review history**

The review history is available as Additional file 11.

**Peer review information**

**Authors' contributions**

FW and JY conceived the project, designed the research, and supervised the experiments; YR and WL acquired data; YH analyzed the sequencing data; MH, JY, SL, XW, YM, YS, HZ, YG, and HZ analyzed and interpreted data; YR, YH, and FW collected and assembled data; YR and YH drafted the manuscript; SR, JW, and LX revised the manuscript for important intellectual content; JW, XW, JY, and FW obtained funding. All author(s) read and approved the final manuscript.

**Funding**

This work was supported by the National Key Research and Development Program of China [2019YFA0111700 to X.W.] and [2019YFA0801800 to J.Y.]; CAMS Innovation Fund for Medical Sciences [2018-I2M-1-001 to Y.R. and X.Z., 2019-I2M-2-001 to J.Y., 2017-I2M-3-009 to X.W. and J.Y., 2016-I2M-3-002 to F.W., and 2017-I2M-1-015 to H.Z.]; the National Natural Science Foundation of China [81530007 to J.Y., 31725013 to J.Y., 82022001 to F.W. and 81970103 to F.W.]; the Fundamental Research Funds for the Core facility [3332019001], the CAMS [2016GH310001 to J.Y., 2017-I2M-B&R-04 to J.Y. and 2018RC310013 to F.W.] and Grant from Medical Epigenetics Research Center, CAMS [2017PT31035]. Funding for open access charge: National Key Research and Development Program of China [2019YFA0801800 to J.Y.].

**Availability of data and materials**

The ChIP-seq datasets of histone modifications and Pol II were taken from NCBI BioProject ID: PRJNA510375(H3K27ac ChIP-seq datasets) [100], PRJNA295216(H3K4me3 ChIP-seq datasets) [101], PRJNA63443 (H3K27me3, H3K79me2, H3K36me3 ChIP-seq datasets) [102], PRJNA382164(Pol II ChIP-seq datasets) [103]. The expression profiling after QKI5 knockdown in THP-1 cells from GSE74887 [10].

Raw sequencing and processed datasets have been deposited in the Gene Expression Omnibus (GEO) with the accession number GSE161943 [104]. Details of these data sources are listed in Additional file 10: Table S9. Additional file 3 and Additional file 8 contain MS data on all peptides and proteins identified in this study. The mass spectrometry datasets of RBPs screening in THP-1, 293T, and K562 cells were available at PRIDE website with identifiers PXD028787 [105], PXD028660 [106], and PXD028668 [107]. The mass spectrometry (MS) datasets of QKI5 co-immunoprecipitation samples in THP-1 cells were available at PRIDE website with identifiers PXD028743 [108]. Details of these data sources are listed in Additional file 10: Table S9.

# Declarations

**Ethics approval and consent to participate**

Not applicable

**Consent for publication**

Not applicable

**Competing interests**

The authors declare that they have no competing interests.

**Author details**

¹State Key Laboratory of Medical Molecular Biology, Department of Biochemistry and Molecular Biology, Institute of Basic Medical Sciences, Chinese Academy of Medical Sciences, School of Basic Medicine Peking Union Medical College, Beijing 100005, China. ²Key Laboratory of RNA and Hematopoietic Regulation, Chinese Academy of Medical Sciences, Beijing 100005, China. ³Department of Pathophysiology, State Key Laboratory of Medical Molecular Biology, Peking Union Medical College,  Beijing 100005, China. ⁴Emergency Department of West China Hospital, Sichuan University, Chengdu 610014, China. ⁵Department of Thoracic Surgery, Nanfang Hospital, Southern Medical University, Guangzhou 510515, China. ⁶Medical Epigenetic Research Center, Chinese Academy of Medical Sciences, Beijing 100005, China.

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

## 

