## [**Additional file 11.** Review history · Genome Biology]

Review History

First round of review

Reviewer 1

Are you able to assess all statistics in the manuscript, including the appropriateness of statistical tests used? No, I do not feel adequately qualified to assess the statistics.

Comments to author:

Title: A global screening identifies chromatin-enriched RNA-binding proteins and the transcriptional regulatory activity of QKI5 during monocytic differentiation

Comments: This paper starts with identification of chromatin-associated RNA-binding proteins and then focuses on a representative RBP—QKI5 to reveal its functional role during monocytic differentiation. The authors show that QKI5 binds to the promoters of many differentiation-associated genes, in a manner independent of the RNA-binding activity of QKI5. Depletion or overexpression of QKI5 led to impaired differentiation. These observations are potentially interesting. However, the experimental data fall short to support their current conclusions. A major revision should be made in order to publish this work.

Major Concerns:

1. The authors concluded that the role of QKI on chromatin is independent of its RNA-binding activity, based on the assays using an RRM mutant of QKI. And RNase treatment failed to affect QKI mutant's chromatin association as shown in Figure S3h. Yet, this conclusion seems to be contradictory to the chromatin fractionation assays. As shown in Fig.2b, RNase treatment led to the decreased association of QKI on chromatin, which appeared to be consistent among different cell types. In addition to QKI5, the binding of ADAR and ELAVL1 to chromatin seemed to partially rely on RNA. The author needs to provide explanation for this discrepancy.

2. The authors performed ChIP-seq and CLIP-seq for several RBPs. But the data mining is far from enough. Here are my questions and suggestions.

(1) What kinds of genes are bound by these RBPs at the DNA and/or RNA levels? Do they tend to bind house-keeping genes or cell-type specific genes?

(2) What's the relationship between ChIP-seq and CLIP-seq for each RBP. Besides correlation analysis at the gene level, the authors should provide more sequencing tracks or peak-level analysis. And how is the association of these two datasets related to their sensitivity to RNase treatment?

(3) ChIP-seq and CLIP-seq of these RBPs showed quite different binding patterns and had only small fractions of overlapped genes. The correlation index for ChIP-seq and CLIP-seq is also low for each RBP analyzed. Could it result from mRNA contamination in CLIP-seq? Do these RBPs possess both DNA- and RNA-binding capacities? Do they have RNA-dependent and independent functions?

(4) What about meta-plot pattern at the gene body of QKI5 ChIP targets compare to the input? The enrichment of QKI5 ChIP-seq signals at the promoter is not sufficient to indicate a regulatory role on gene transcription. What about Pol II ChIP-seq or H3K4me3 ChIP-seq signals on commonly-activated genes after QKI5 knockdown?

3. The authors claimed that QKI promotes monocyte differentiation. Based on Figure S3a, the expression of QKI5 is increased on day 8 but recovered to the normal expression level on day 11 of differentiation in HSPCs and THP-1 cells. However, QKI5 KD or OE affected monocyte differentiation on D13 and D19 in HSPCs. The authors should check the phenotypes at D8 instead of later time points when QKI5's expression is decreased.

4. Based on ChIP-seq analysis, QKI binds to >1000 gene promoters. But the authors only focused on a small number of targets that are related to differentiation. How about other target genes?

5. EMSA detected weak supershifted signals between QKI and dsDNA probes. This weak DNA-binding affinity of QKI hardly explains its chromatin-binding. More control DNA probes with different sequences (such as the common motif of QKI5 ChIP targets) should be tested. In addition, the authors should compare the dsDNA-binding activity of QKI to its abilities to bind ssDNA, DNA/RNA hybrid, and RNA.

Minor Concerns:

1. In the first part, the authors mentioned an observation that RNase treatment had different effects on RBP's chromatin associations in different cell lines, which indicates a context-dependent regulation. Yet it's a pity that the authors didn't extend this part or discuss the potential mechanism.

2. The authors characterized QKI in-depth based on an odds ratio (OR) calculation. But QKI5's OR is a slightly higher than 1 (Fig. 2g). Please explain the logic to choose QKI.

3. The authors used RBPDB and ATtRACT to define the RBP library, which clearly does not include all the RBPs. There are other resources for RBPs (for example, Tuschl, 2014).

4. Based on Figure S1h, RNAs were not fully digested.

Reviewer 2

Are you able to assess all statistics in the manuscript, including the appropriateness of statistical tests used? Yes, and I have assessed the statistics in my report.

Comments to author:

A global screening identifies chromatin-enriched RNA-binding proteins and the transcriptional regulatory activity of QKI5 during monocytic differentiation

by Ren et al.

Ren et al. perform a systematic proteomics screen to identify RNA binding proteins that are associated with the chromatin in three different cell lines. Among the candidate list they decide to focus on seven proteins that have been linked to haematopoiesis. For these proteins they perform genome-wide studies on RNA and DNA binding. From these experiments they find that QKI5 is binding to promoter regions. There it acts independent of its role as RNA binding protein and controls critical monocytic differentiation-associated

genes. The overall topic of the interplay between Chromatin and RNA biology is very relevant and the findings are interesting. However, in the current state the quality of the genome-wide datasets and analysis are difficult to judge and will require some additional experiments and analyses before publication. For more details see comments below.

Comments:

1) The authors perform CLIP experiments for several RBPs. However, it is difficult to evaluate the quality of these experiments. This is of particular interest for QKI5 CLIP since the authors report binding patterns that contradict previous studies. Are the antibodies used for Immunoprecipitation (IP) specific to QKI5? The authors should show that for the different IPs the protein of interest is pulled down and no contaminations. For QKI5 it would be required to perform the CLIP IP in a QKI5 knockdown/knockout background to show that the purified QKI5-RNA complexes are lost.

Also, it would be required to report some statistics in the methods section: How many reads remained per sample after mapping. Has PCR duplicate removal been performed? How many replicates have been performed (At least 2-3 would be required!)? How many binding sites are there for the different proteins? Are they reproducible between the replicates?

2) Considerations from comment 1 also hold true for the Chip-seq experiments. For example, in Figure 5C it looks like the read coverage for the QKI5 Chip is very low, with 5 or 10 reads forming a peak. Have these experiments been performed in replicates? This would be required and should be shown for some examples, at least as supp. Figure.

3) The question if RBP binding leads to changes in transcript abundance upon knockdown of the RBP is very interesting. However, I am not convinced by the current analysis. To make it more convincing it would be required to make use of the dataset as a whole. Since the authors have RBP binding data for seven proteins as well as knockdown data for the same proteins they should compare all by all. Meaning calculating the odds ratio of all RBP binding sets versus all knockdown effects. This would be a nice internal control to show that the observed effects are specific.

Authors Response

Point-by-point responses to the reviewers' comments:

Referee 1:

Comments: This paper starts with identification of chromatin-associated RNA-binding proteins and then focuses on a representative RBP—QKI5 to reveal its functional role during monocytic differentiation. The authors show that QKI5 binds to the promoters of many differentiation-associated genes, in a manner independent of the RNA-binding activity of QKI5. Depletion or overexpression of QKI5 led to impaired differentiation. These observations are potentially interesting. However, the experimental data fall short to support their current conclusions. A major revision should be made in order to publish this work.

Our response: We thank this reviewer for the comments and advices. We have carefully revised our manuscript according to the reviewer's suggestion, in principle, we have (i) added more information and description in both results and methods section; (ii) provided detailed explanation to each relevant question, for instance, we have performed more comparative analyses between RBPs' ChIP-seq and CLIP-seq results and tried to make a conclusive speculation of the regulatory pattern of these Che-RBPs as well as more intensive analyses of QKI5 ChIP-seq data along with other data to provide more supporting evidences for the potential transcriptional activation ability of QKI5; (iii) included some essential quality control experiments were to support the main findings.

Major Concerns:

1. The authors concluded that the role of QKI on chromatin is independent of its RNA-binding activity, based on the assays using an RRM mutant of QKI. And RNase treatment failed to affect QKI mutant's chromatin association as shown in Figure S3h. Yet, this conclusion seems to be contradictory to the chromatin fractionation assays. As shown in Fig.2b, RNase treatment led to the decreased association of QKI on chromatin, which appeared to be consistent among different cell types. In addition to QKI5, the binding of ADAR and ELAVL1 to chromatin seemed to partially rely on RNA. The author needs to provide explanation for this discrepancy.

Our Response: Thanks for the comments. In Fig. 2b (Supported file 1-1: a), RNase treatment induced a partial dissociation of chromatin-bound QKI5 indicating that the interaction between QKI5 and chromatin were depended on RNA to some extent, whereas Fig. S3h (Supported file 1-1: b) showed that QKI5 and QKI5 mutant bound chromatin with comparable intensity, which indicated the RNA-independent chromatinbinding ability of QKI5. This inconsistency is probably due to the experimental design of RNase treatment assay. The RNase A treatment was added in the last step as described in the method: "chromatin fractions were incubated with 20µg/ml RNase A in PBS for 5min at 4°C, while RNase A-free samples were incubated with PBS alone." Although QKI5 mutant might dissociate from chromatin, the unbound part could not be detected in SNE fraction, which was collected before RNase treatment. Therefore, the observation that QKI5 mutant still abounded in chromatin-fraction as shown in Figure S3h was possibly due to the overexpression of QKI5 mutant was higher than wild-type QKI5 as shown in Supported file 1-1: c. As for ADAR and ELAVL1, their binding to chromatin is indeed dependent on RNA partially, as the existence of RNA molecule strengthened the interaction between this type of RBP and chromatin. The purpose of Fig. 2b (Supported file 1-1: a) is to demonstrate the

marked variation of RNA-dependency of RBPs' association with chromatin, namely, different RBPs show different RNA-dependency when interact with chromatin, suggesting different interaction patterns between RBPs and chromatin.

2. The authors performed ChIP-seq and CLIP-seq for several RBPs. But the data mining is far from enough. Here are my questions and suggestions.

(1) What kinds of genes are bound by these RBPs at the DNA and/or RNA levels? Do they tend to bind house-keeping genes or cell-type specific genes?

Our response: According to the reviewer's suggestion, we first analyzed the distribution of each hChe-RBP in different gene types, which indicated that hChe-RBPs exhibited dissimilar binding patterns in different types of genes either at DNA or RNA level. As shown in Fig. 2c, ChIP-seq analyses revealed that each hChe-RBP possessed a unique distribution pattern in genome, yet all presenting a trend to bind protein-coding genes, while PTBP3 showed binding preference to lncRNA genes (Supported file 1-2: a, left panel, see "Methods" for detail). Additionally, CLIP-seq results indicated that hChe-RBPs preferred to bind to protein-coding RNA transcripts, among which NUDT21 also showed a tendency to bind small RNA transcripts. The enrichment level of each hChe-RBP, evaluated by fold-change between IP and Input samples, also differed among different gene types at DNA or RNA levels (Supported file 1-2: b). For example, QKI5 tended to accumulate on genomic regions of both lncRNA and small RNA but only showed distinct enrichment on RNA transcripts of small RNA. NUDT21 exhibited high binding-intensity on RNA transcripts of small RNA but showed no gene-type specific binding-tendency on genome. Subsequently, besides distribution pattern analysis of gene types, we also analyzed different types of protein-coding genes from hChe-RBPs' ChIP-seq and CLIP-seq results. According to the reviewer's advice, we divided protein-coding genes into house-keeping (HK) genes and cell-type specific (SP) genes in THP-1 cells, acquired from the human protein atlas database (<https://www.proteinatlas.org/>). For each hCheRBP, we calculated the ratio of HK and SP genes in the RBP-bound genes at DNA or RNA level and took the ratio of HK or SP genes of THP-1 cells in overall coding-genes as the reference. The binding tendency was determined by comparing the proportion of HK or SP genes in each hChe-RBP's bound genes with the reference ratio. For ChIP-bound genes: hChe-RBPs possessed different binding tendencies towards HK genes while showed no preference to SP genes in THP-1 cells, as ADAR, KHSRP, ELAVL1 and SETD1A preferred to bind HK genes. Of note, SETD1A, a known methyltransferase which methylates histone H3 lysine 4 (H3K4) (Wysocka J, et. al., Genes Dev, 2003,17(7):896-911), exhibited the highest percentage (38.7%) of HK gene binding. For CLIP-bound genes: all these RBPs showed different degree of preference for HK genes. However, ELAVL1 also showed tendency to bind SP genes in THP-1 cells suggesting its specific regulatory role in cell-type specific pathways (Supported file 1-2: c).

(2) What's the relationship between ChIP-seq and CLIP-seq for each RBP. Besides correlation analysis at the gene level, the authors should provide more sequencing tracks or peak-level analysis. And how is the association of these two datasets related to their sensitivity to RNase treatment?

Our response: As the reviewer suggested, in order to explore the interaction between hChe-RBPs' DNA and RNA binding, we analyzed the relative positions between ChIPseq and CLIP-seq peaks for each hChe-RBP: For each hChe-RBP, the overlapping rate between

ChIP and CLIP peaks was very low, with only 0.14% ChIP peaks overlapped with CLIP peaks in average (Supported file 1-3: a). CLIP peaks only appeared concurrently in the range of 5kb~10kb or beyond 10kb of ChIP peaks (~17% and 80% in average, respectively), corresponding to the low overlapping rate at gene level presented by Jaccard index (Supported file 1-3: b). We speculated that the interaction of hChe-RBPs to chromatin might not be mediated by newly-transcribed RNAs, nor by neighbor nascent transcripts because the distances between these RBPs' DNA and RNA binding sites are mostly more than 5kb. Their interaction might be mediated by ncRNAs like LncRNAs which could function at long distance or by the high-ordered structures of chromatin.

Combining with RNase treated sub-cellular fractionation results in THP-1 cells (Supported file 1-4: a, b), we could speculate the interaction modes of hChe-RBPs as: 1. DNA/RNA separated, e.g. SETD1A; 2. DNA/RNA combined, e.g. NUDT21/QKI5/PTBP3/ADAR/KHSRP/ELAVL1, as summarized in the schematic diagram (Supported file 1-4: d). From ChIP/CLIP-seq comparative analysis (Supported file 1-4: c), SETD1A bound DNA more than RNA at both peak level and gene level, and it also showed the highest DNA-binding rate among all hChe-RBPs tested in our study (Supported file 1-4: c), indicating its preference to interact with chromatin, which was in accordance with RNase treated sub-cellular fractionation results (Supported file 1-4: a). In addition, the odd ratio calculated by comparing hChe-RBP-binding genes and differentially expressed genes (DEGs) induced by RBP knock-down also showed that SETD1A possessed the strongest transcriptional regulation potential (Supported file 1-4: e). Other hChe-RBPs' interactions with chromatin relied on RNA at different degrees as shown in RNase treatment assay (Supported file 1-4: a). In summary, our results suggest that the hChe-RBPs possess DNA/RNA separated or combined chromatin interaction mode accord with their different DNA- or RNA-binding tendency. The separated chromatin and RNA interaction of RBPs indicates that they might have unique and unrelated regulatory function at DNA or RNA level.

(3) ChIP-seq and CLIP-seq of these RBPs showed quite different binding patterns and had only small fractions of overlapped genes. The correlation index for ChIP-seq and CLIP-seq is also low for each RBP analyzed. Could it result from mRNA contamination in CLIP-seq? Do these RBPs possess both DNA- and RNA-binding capacities? Do they have RNA-dependent and independent functions?

Our response: We thank the reviewer for this concern. Actually, with the high specificity of CLIP experiment, the probability of mRNA contamination is quite low. According to CLIP-antibody quality control (Supported file 1-5: a) and reproductive test of CLIP-seq data (Supported file 1-5: b, right panel), the experiment process and the quality of data were standardized. To further address the reviewer's concern, we have compared the known RNA motifs of classical RBPs as QKI5, ADAR, KHSRP, PTBP3 and ELAVL1 with our CLIP-seq data, which showed high consistency between those identified motifs generated from our data and the published ones, indicating that the CLIP signals in our data possessed high specificity and accuracy (Supported file 1-5: c). As showing below (Supported file 1-5: c), 5 RBPs' (QKI5, ADAR, KHSRP, PTBP3, ELAVL1) CLIP-seq motifs were consistent with those previous studies (Eggington JM. et al. Nat Commun. 2011, 2, 319; Galarneau A. et al., Nat Struct Mol Biol, 2005,12(8), 691-8; Van Nostrand EL. et al., Nature, 2020, 583(7818), 711-719; Meisner, N.-C., et al., ChemBioChem, 2004, 5: 1432-1447).

According to our data, besides RBP's classical functions in RNA regulation, they might have transcriptional regulation potential. Among them, QKI5, KHSRP and SETD1A tended to

regulate gene transcription by binding to target genes (Supported file 1-6). In addition, SETD1A is a known methyltransferase possessing DNA binding domain (Wysocka J, et. al., *Genes Dev*, 2003,17(7):896-911); KHSRP and ADAR have also been reported to have DNA binding domains (Davis-Smyth T. et.al., *J Biol Chem*. 1996, 271(49):31679-87; Schwartz T, et al. *Science*. 1999, 284(5421):1841-5); as for QKI5, although recent studies have shown that it could be recruited to chromatin by transcription factors (Zhou X, et.al., *J Clin Invest*. 2020,130(5):2220-2236; Shin S, et.al., *Nat Commun*. 2021,12(1):3005), there's still no study on whether QKI5 possesses DNA binding domain or not. Furthermore, other RBPs tested in our study haven't been shown to contain any known DNA binding domains previously. As described in our manuscript, RBPs could be multi-functioning factors regulating gene expression at different layers. And for transcriptional regulation, they might function by direct (with DNA binding motif) or indirect (without known DNA binding domain, coordinate with other factors) interactions with chromatin.

(4) What about meta-plot pattern at the gene body of QKI5 ChIP targets compare to the input? The enrichment of QKI5 ChIP-seq signals at the promoter is not sufficient to indicate a regulatory role on gene transcription. What about Pol II ChIP-seq or H3K4me3 ChIP-seq signals on commonly-activated genes after QKI5 knockdown?

Our response: QKI5 preferred to enrich on gene promoters than gene body regions as shown below (Supported file 1-7: a). However, the enrichment on promoters was still low, which was probably due to the weak interaction between QKI5 and chromatin. As the reviewer mentioned, the enrichment on promoters was insufficient to indicate a transcriptional activation ability. To address this concern, we firstly analyzed the histone modifications around QKI5 ChIP peaks and found that the active histone markers as H3K4me3/H3K27ac tended to enrich on QKI5-located promoters indicating its transcriptional activation function. At the same time, H3K27me3, a repressive histone maker was relatively low at QKI5-located promoters (Supported file 1-7: b). Moreover, we performed H3K4me3 and Pol II ChIP-seq in QKI5 knock-downed (QKI5 KD) THP-1 and control cells, respectively. Our results showed that both H3K4me3 and Pol II signals decreased on QKI5 commonly-activated genes in QKI5 KD group, providing further evidence on QKI5's transcriptional activation function (Supported file 1-7: c). In addition, loss of QKI5 only brought minor decreases of H3K4me3 and Pol II ChIP signals. This is probably because that QKI5 is not a general transcription factor or a histone modification enzyme so that the deficiency of QKI5 would not lead to broad or significant impact on the enrichment of H3K4me3 and Pol II, i.e., the chromatin environment.

3. The authors claimed that QKI promotes monocyte differentiation. Based on Figure S3a, the expression of QKI5 is increased on day 8 but recovered to the normal expression level on day 11 of differentiation in HSPCs and THP-1 cells. However, QKI5 KD or OE affected monocyte differentiation on D13 and D19 in HSPCs. The authors should check the phenotypes at D8 instead of later time points when QKI5's expression is decreased.

Our response: Thanks for the suggestion. We had checked the influence of QKI5 on early time point during HSC monocytic differentiation (day 8), and QKI5 also showed monocytic-differentiation promoting effect at the indicated time point. As shown below, QKI5 and its mutant could promote monocytic differentiation, while knock-down of QKI5 impeded the same process (Supported file 1-8).

4. Based on ChIP-seq analysis, QKI binds to >1000 gene promoters. But the authors only focused on a small number of targets that are related to differentiation. How about other target genes?

Response: According to our QKI5 ChIP-seq data, although QKI5 locates on many gene promoters, it doesn't mean that the expression of these genes is regulated by QKI5. By combining RNA-seq profile with biological pathway screening, we could find the functional target genes regulated by QKI5. Based on our group's previous findings (Wang F. et al., Cell Res, 2017, 27:416-439; e Bruin RG. et al., Nat Commun, 2016, 7:10846; Fu H. et al., Mol Biol Cell, 2012, 23:1628-1635), we were particularly interested with hematopoiesis-related genes. Besides hematopoiesis, GO result showed that QKI5 preferred to locate on genes involved in neurodevelopment, cardiac development, and bone development according to previous reports (Supported file 1-9: a) (Hayakawa-Yano Y. et al., Genes Dev, 2017, 31(18), 1910-1925; Hayakawa-Yano Y. et al., Int J Mol Sci, 2019, 20(5), 1010; Rauwel B. et al., J Bone Miner Res, 2020, 35(4), 753-765; Gupta SK. et al., Circ Res, 2018, 122(2), 246-254; Guo W. et al., Cell Physiol Biochem, 2011, 28(4), 593-602). Moreover, we also performed GO analysis of the overlaps between QKI5-bound and QKI5-regulated genes. The result showed that these QKI5 transcriptionally-regulated genes tended to enrich in myeloid leukocyte-related pathways confirming our conclusions (Supported file 1-9: b, Additional file 7: Table S6).

5. EMSA detected weak supershifted signals between QKI and dsDNA probes. This weak DNA-binding affinity of QKI hardly explains its chromatin-binding. More control DNA probes with different sequences (such as the common motif of QKI5 ChIP targets) should be tested. In addition, the authors should compare the dsDNA-binding activity of QKI to its abilities to bind ssDNA, DNA/RNA hybrid, and RNA.

Our response: We thank the reviewer for suggesting this important control experiment. According to the reviewer's suggestion, we have generated ssDNA, DNA/RNA hybrid, and RNA probes with the same sequence of previously used CXCL2 promoter probe as well as QKI5 ChIP motif probe (two motif sequence tandemly connected). The result of EMSA assay using these probes showed that QKI5 had the strongest interaction with RNA probes, which was in accordance with its RBP identity. QKI5 also showed weak interaction with CXCL2 promoter probe (dsDNA) which was consistent with our previous finding. And QKI5 might interact with DNA/RNA hybrid probe reflected in the slight tailing emerged in the indicated lane (Supported file 1-10: a). In summary, QKI5 binds RNA with high intensity while shows low binding intensity with dsDNA, which is possibly because of that none of known DNA-binding domain existed in QKI5 protein.

However, there might be uncharacterized DNA binding structures, or the weak DNA binding feature of some known structures in QKI5 might not be recognized so far, which resulted the weak bands bound to QKI5 protein in the EMSA analysis. In addition, QKI5 could also interact with the ChIP motif probe, further demonstrating that QKI5 could bind DNA by recognizing specific sequence (Supported file 1-10: b).

Minor Concerns:

1. In the first part, the authors mentioned an observation that RNase treatment had different effects on RBP's chromatin associations in different cell lines, which indicates a context-

dependent regulation. Yet it's a pity that the authors didn't extend this part or discuss the potential mechanism.

Our response: Thanks for the comments. Please refer to our response to Major concerns 2/Section 2 and 3 for more details as well as the discussion part in the revised manuscript from line 420 to 432.

2. The authors characterized QKI in-depth based on an odds ratio (OR) calculation. But QKI's OR is a slightly higher than 1 (Fig. 2g). Please explain the logic to choose QKI.

Our response: We apologize for the unclear description of why choose QKI. The logic or the filtering process to pick up QKI is: First, the overlapping rate of QKI's ChIP/CLIP signals is relatively low suggesting it might possess unique regulatory function at the transcriptional level (Fig.2g, ADAR/QKI5/PTBP3/NUDT21/ELAVL1 possessed low ChIP/CLIP-gene co-occupancy rate). Next, histone markers labeling transcriptional active status enrich on QKI5-located promoter regions that indicates its transcriptional activation role (Fig.2i, the promoter regions of ADAR/QKI5/KHSRP/SETD1A were enriched of active histone markers like H3K4me3 and H3K27ac. The candidates after the two-step filtration are only ADAR and QKI5). Finally, according to Fig.2j, the transcriptional regulatory potential of each RBP calculated by OR shows that QKI5 might regulate target-genes' expression at transcriptional level (Fig.2j, QKI5/KHSRP/SETD1A showed significant transcriptional regulatory potential presented by OR ratio. The only candidate after the final filtration is QKI5). In addition, the target genes regulated by QKI5 gathered in hematopoietic related pathways the most among other hChe-RBPs with transcriptional regulatory potential (Fig. 2k), thus making QKI5 a potentially functional hChe-RBP in hematopoiesis.

3. The authors used RBPDB and ATtRACT to define the RBP library, which clearly does not include all the RBPs. There are other resources for RBPs (for example, Tuschl, 2014).

Our response: RBPDB (<http://rbpdb.ccbr.utoronto.ca/>) is a database focusing on the collection of experimentally-validated RBPs and RNA binding domains (RBDs), and ATtRACT (<http://attract.cnice.es>) contains information on 370 hand-curated and experimentally-validated RBPs associated with 1583 co-nonsense motifs. Together, these databases represent the resource for reliable RBP information with experimental validations, and have been widely accepted in the RBP research community, including in "Ray, D. et al., Nature, 2013, 499, 172–177", "Pierre C. et. al., Cell, 2012, 150, 5, 1068-1081", "Daniel D. et al., Molecular Cell, 2018, 70, 854–867", "Basak E. et al., Mol Syst Biol. ,2019, 15, e8513" and "Girolamo G., et.al., Database, 2016, Volume 2016, baw035".

As the reviewer mentioned, the paper published by Tuschl et al in 2014 reported a census of 1,542 RBPs, which contained RNA-related RBDs identified by hidden Markov models with RNA-related functions. This work predicted a large number of potential RBPs with RBD, but direct experimental evidence of their interaction with RNA was not provided. And other RBPs' databases were based on CLIP-seq datasets, for example starBase (<http://starbase.sysu.edu.cn/index.php>) and POSTAR2 (<http://lulab.life.tsinghua.edu.cn/postar/index.php>) only focus on a small fraction of RBP (36 and 171, respectively). Therefore, we chose RBPDB and ATtRACT databases to define the RBP library with consideration of coverage as well as reliability.

4. Based on Figure S1h, RNAs were not fully digested.

Our response: According to this figure, the specific bands of 18S and 28S RNA disappeared upon RNase treatment indicating the intact RNA molecules were fully degraded. The small fragments of RNA would not completely disappear thus resulting in smears in the lane. In this situation, the intact structure of RNA molecules no longer existed, resulting in the loss of interaction activity between RBP and RNA.

Referee 2:

A global screening identifies chromatin-enriched RNA-binding proteins and the transcriptional regulatory activity of QKI5 during monocytic differentiation

by Ren et al.

Ren et al. perform a systematic proteomics screen to identify RNA binding proteins that are associated with the chromatin in three different cell lines. Among the candidate list they decide to focus on seven proteins that have been linked to haematopoiesis. For these proteins they perform genome-wide studies on RNA and DNA binding. From these experiments they find that QKI5 is binding to promoter regions. There it acts independent of its role as RNA binding protein and controls critical monocytic differentiation-associated genes. The overall topic of the interplay between Chromatin and RNA biology is very relevant and the findings are interesting. However, in the current state the quality of the genome-wide datasets and analysis are difficult to judge and will require some additional experiments and analyses before publication. For more details see comments below.

Our response: Thanks for the comments and suggestions. We added necessary controls in key experiments according to the reviewer's advices and had responded to all the questions as follows.

Comments:

1) The authors perform CLIP experiments for several RBPs. However, it is difficult to evaluate the quality of these experiments. This is of particular interest for QKI5 CLIP since the authors report binding patterns that contradict previous studies. Are the antibodies used for Immunoprecipitation (IP) specific to QKI5? The authors should show that for the different IPs the protein of interest is pulled down and no contaminations. For QKI5 it would be required to perform the CLIP IP in a QKI5 knockdown/knockout background to show that the purified QKI5-RNA complexes are lost.

Response: We thank the reviewer for this suggestion. We have added IP controls for antibodies we used in ChIP/CLIP-seq and results showed that the antibodies could specifically enrich the indicated proteins (Supported file 2-1).

As for quality controls of QKI5 CLIP-seq, we conducted CLIP-seq in QKI5 knockdown THP-1 cells as the reviewer suggested. First, QKI5 protein was successfully knocked down as shown below (Supported file 2-2: a). Next, the CLIP-IP result showed that QKI5 in control

group was immunoprecipitated by the antibody while rarely been immunoprecipitated in QKI5 knockdown (QKI5 KD) group (Supported file 2-2: b). Furthermore, we analyzed the CLIP-seq data of control/QKI5 KD groups respectively, which showed that QKI5 CLIP peaks acquired in control group were much more than that from QKI5 KD group (7337 vs. 669), demonstrating that RNAs bound by QKI5 could hardly be immunoprecipitated after the loss of QKI5 (Supported file 2-2: c). Besides, the overlapped peaks of the two datasets were very few, indicating the inconsistency between the 2 CLIP-seq datasets (Supported file 2-2: d), also the enrichment level of non-overlapped peaks from control group was higher than the overlapped peaks (Supported file 2-2: e). In addition, the read counts in control group were more than that in QKI5 KD group (Supported file 2-2: f). The IGV results also demonstrated that the CLIP peaks on target sites disappeared after QKI5 knock-down (Supported file 2-2: g). Since there's very little RNA could be enriched in QKI5 KD group, we concluded that the QKI5 CLIP peaks we have obtained were specific.

Also, it would be required to report some statistics in the methods section: How many reads remained per sample after mapping. Has PCR duplicate removal been performed? How many replicates have been performed (At least 2-3 would be required!)? How many binding sites are there for the different proteins? Are they reproducible between the replicates?

Our response: We have included the details for statistical analysis of eCLIP-seq data processing in Additional file 10: Table S9, each eCLIP-seq dataset had more than 100 million unique mapping clean reads for downstream analysis. For the eChIP-Seq experiment, we performed two replicates for each RBP. Due to the relatively strong correlation between the replicates (Additional file 1: Figure S2b, right), the two replicates were combined to perform peak finding and downstream data analysis. And we have included detailed data processing of eCLIP-seq in the Method section as described below:

The hChe-RBPs CLIP-seq datasets were processed in accordance with previous studies (Van Nostrand EL, et al., Nat Methods, 2016, 13:508-514), and the eCLIP-seq data processing pipeline was available at "<https://github.com/YeoLab/eclip>". The Raw reads with distinct inline barcodes were demultiplexed using in-house scripts, and the 10-mer random sequence was appended to the reads name in bam file for later usage. Low quality reads and adapter sequence were trimmed by cutadapt (Martin M., et al., J EMBnet.journal, 2011, 17:3). Repetitive reads were removed by aligning reads with human repetitive element sequence on RepBase database (<https://www.girinst.org/>) by STAR. Cleaned reads were mapped to Homo sapiens genome (Ensembl GRCh38.p5) by STAR (Dobin A., et al., Bioinformatics, 2013, 29:15-21). PCR duplicate reads were removed by in-house script based on sharing identical random sequence. Two biological replicates were merged by SAMtools "merge" for following analysis. The CLIP-seq data processing statistics were list in Additional file 10: Table S9. Peaks calling and downstream data analysis were performed using clipper (Lovci MT., et al., Nat Struct Mol Biol, 2013, 20:1434-1442) software. Peaks normalization by "Peak_input_normalization_wrapper.pl" tools, available at "<https://github.com/YeoLab/eclip>". The CLIP-seq peaks were filtered by $P\text{-value} < 10e-3$ and $\text{fold-change} > 4$. Enrichment of hChe-RBP's RNA binding sites on the human genomic region was calculated by R package (ChIPseeker) (Yu G., et al., ChIP seeker: An R/Bioconductor package for ChIP peak annotation, comparison and visualization, 2015). The repeatability between two biological replicates was evaluated by Pearson correlation coefficient with read coverages for genomic regions per 1000bp, which were generated from Deeptools "multiBamSummary" (Ramírez F., et al., Nucleic Acids Res, 2016, 44:W160-165).

2) Considerations from comment 1 also hold true for the Chip-seq experiments. For example, in Figure 5C it looks like the read coverage for the QKI5 Chip is very low, with 5 or 10 reads forming a peak. Have these experiments been performed in replicates? This would be required and should be shown for some examples, at least as supp. Figure.

Our response: We are confident that the ChIP-seq signals in our study were highly specific and reliable confirmed by both antibody-specificity verification and normative data-processing process with 2 replicates.

For the ChIP-Seq experiment, we performed two replicates for each RBP. The repeatability between two biological replicates was evaluated by Pearson correlation coefficient of reads coverage for genomic regions per 1000bp, which generated from Homer “getPeakTags”. With the relatively strong correlation between the replicates (Supported file 2-3: a, left), the two replicates were pooled to perform peak calling and downstream data analysis.

For the peak visualization, we inputted a “bigwig” format file, which had been normalized the total number of reads to 1×10^7 for comparing different samples. The y-axis of IGV-plot represented the ChIP-fragment density, which was defined as the total number of overlapping fragments at each position in the genome. The ChIP fragment density was low because of the normalization methods, and the values represent relative reads coverage for comparing different samples rather than absolute reads coverage. We had provided the IGV-plot contained actual reads of binding sites on promoter regions of MOV10 and UACA genes which were QKI5 ChIP target genes verified in our work (Supported file 2-3: b).

3) The question if RBP binding leads to changes in transcript abundance upon knockdown of the RBP is very interesting. However, I am not convinced by the current analysis. To make it more convincing it would be required to make use of the dataset as a whole. Since the authors have RBP binding data for seven proteins as well as knockdown data for the same proteins they should compare all by all. Meaning calculating the odds ratio of all RBP binding sets versus all knockdown effects. This would be a nice internal control to show that the observed effects are specific.

Our response: Thank this reviewer for the advice. In the revised manuscript, we had calculated the odds ratio of all hChe-RBPs' binding genes vs. all differentially expressed genes (DEGs) generated by hChe-RBPs' knockdown effects. But the OR of all-by-all sets was 4.38 (p -value $< 2.2 \times 10^{-16}$) which indicated a strong relevance between the two datasets (data not shown). We realized that the 7 tested hChe-RBPs were screened out with hematopoiesis-related functions thus their target genes might be partially overlapped causing functional relevance. And in this case, the OR of all-by-all sets was unsuitable to be the internal control. To test this hypothesis, we analyzed the overlapping ratio of the 7 RBPs' bound genes or DEGs upon each knockdown effect. The results showed that the pairwise overlapping ratios between each 2 of these 7 RBPs' ChIP-target genes could reach up to 0.77, meanwhile the overlapping ratios could be as high as 0.37 at DEG level (Supported file 2-4: a), indicating high overlapping between ChIP-binding sets and DEG sets which implied the functional relevance. Especially, SETD1A, also known as a histone methyltransferase, distributed broadly on genome covering up to 41% (17144/41480) of expressed genes in THP-1 cells (Supported file 2-4: a), thus overlapping largely with other RBPs' bound genes. Therefore, due to the high overlapping rates between SETD1A with other hChe-RBPs as an example, it is inappropriate to use this all-by-all strategy as the internal control. To strengthen the reliability of our analyses, we applied an IgG ChIP-seq dataset generated from wildtype

THP-1 cells as a randomized control representing the background noise of non-specific binding signals which was irrelevant to the DEGs. By using this randomized control, the OR of QKI5, KHSRP and SETD1A were higher than the controls (grey columns represent the OR of IgG ChIP-seq vs. RBPs RNA-seq) confirming the correlation between their binding and the expression of target genes (Supported file 2-4: b).

Supported file 1-1 (for reviewer #1 major concern 1)

Supported file 1-1 a Immuno-blot validation of the distribution of hChe-RBPs in SNE and CPE fractions with/without RNase treatment of THP-1 cells. **b** Validation of the sub-cellular fractionation detection of QKI5 in Ctrl- and QKI5/QKI5 M1-overexpressing THP-1 cells. **c** Upper panel: Immune blot of QKI5 in Ctrl- and QKI5/QKI5 M1-overexpressing THP-1 cells used in (**b**). Lower panel: Quantitative analysis of QKI5 level from immune blots in (**c**). Error bars indicate standard deviations around the mean of three experimental replicates. Asterisks indicate a significant difference between the specified samples (*P-value < 0.05, ***P-value < 0.001, t test).

Supported file 1-2 (for reviewer #1 major concern 2-(1))

a (Fig. 2c)

b (Additional file 1: Figure S2e)

c (Additional file 1: Figure S2g)

Supported file 1-2 a hChe-RBPs' distribution frequencies on different types of genes revealed by ChIP-seq (left panel) and CLIP-seq (right panel) datasets. **b** hChe-RBPs' enrichment on different gene types calculated by log₂ FC (IP vs. Input) of ChIP-seq (left panel) and CLIP-seq (right panel) peaks. (FC: fold change). **c** hChe-RBPs' occupation frequencies on DNA or RNA transcripts of house-keeping (HK) genes and cell-type specific (SP) genes in THP-1 cells. Blue bars represent ChIP-seq results and pink bars indicate CLIP-seq results.

Supported file 1-3 (for reviewer #1 major concern 2-(2))

a (Fig. 2f)

b (Fig. 2g)

Supported file 1-3 a Heatmap showing the distribution ratios of CLIP-seq peaks neighboring ChIP-seq peaks of each hChe-RBP in different distance ranges on genome. **b** Comparison of hChe-RBP co-occupied genes from ChIP-seq and CLIP-seq datasets. The x-axis shows the Jaccard index of each hChe-RBP's ChIP-seq and CLIP-seq occupied genes, with bubble size indicating co-occupied gene number.

Supported file 1-4 (for reviewer #1 major concern 2-(2))

Supported file 1-4 a Immuno-blot validation of the distribution of hChe-RBPs in SNE and CPE fractions with/without RNase treatment of THP-1 cells. **b** Comparison of protein level quantified by quantitative analysis of the corresponding immuno-blot results. RNA-dependency of Che-RBPs' deposition on chromatin was calculated as the ratio of:

$\frac{\text{Protein CPE (RNase+)}/\text{Protein CPE (RNase-)}}{\text{Histone H3 CPE (RNase+)}/\text{Histone H3 CPE (RNase-)}} > 1$. RNA-independent Che-RBPs are indicated by pink

bars with $\frac{\text{Protein CPE (RNase+)}/\text{Protein CPE (RNase-)}}{\text{Histone H3 CPE (RNase+)}/\text{Histone H3 CPE (RNase-)}} > 1$. RNA-dependent Che-RBPs are

indicated by blue bars with $\frac{\text{Protein CPE (RNase+)}/\text{Protein CPE (RNase-)}}{\text{Histone H3 CPE (RNase+)}/\text{Histone H3 CPE (RNase-)}} < 1$. **c** Comparison of

occupied peak numbers (upper panel) and gene numbers (lower panel) between ChIP-seq and

CLIP-seq datasets of indicated hChe-RBPs. **d** Schematic diagram representing different

interaction modes of hChe-RBPs to chromatin. **e** Upper panel: Odds ratio of expression levels

determined by RNA-seq upon hChe-RBP knocking down on hChe-RBP-occupied versus non-

occupied genes (*P-value < 0.05, double-tail Fisher's exact test). The odds ratio of IgG ChIP-seq

dataset versus RNA-seq datasets upon hChe-RBPs knock-down was used as a randomized control.

Lower panel: Formula of odds ratio.

Supported file 1-5 (for reviewer #1 major concern 2-(3))

a (Additional file 1: Figure S2b) b (Additional file 1: Figure S2c)

c (Additional file 1: Figure S2d)

Supported file 1-5 a Immuno-blots of hChe-RBPs immunoprecipitated using indicated antibodies. IgG was used as negative control of immunoprecipitation assay. **b** Repeatability test of ChIP-seq (left panel) and CLIP-seq (right panel) data. Pearson correlation coefficient was used to evaluate the repeatability of two biology replicates in ChIP-seq and CLIP-seq datasets. **c** CLIP motif prediction of indicated hChe-RBPs by MEME. The table below shows the reported CLIP motifs of the hChe-RBPs compared with motifs generated from our CLIP datasets.

Supported file 1-6 (for reviewer #1 major concern 2-(3))

(Fig. 2j)

Supported file 1-6 Upper panel: Odds ratio of expression levels determined by RNA-seq upon hChe-RBP knocking down on hChe-RBP-occupied versus non-occupied genes (*P-value < 0.05, double-tail Fisher's exact test). The odds ratio of IgG ChIP-seq dataset versus RNA-seq datasets upon hChe-RBPs knocking down was used as a randomized control. Lower panel: Formula of odds ratio.

Supported file 1-7 (for reviewer #1 major concern 2-(4))

Supported file 1-7 a Metaplot showing the distribution of QKI5 ChIP-seq fragment depth within -3000bp to 3000 bp centered around QKI5-target gene regions. The fragment depth was calculated by Deeptools software, which subtract by ChIP-seq Input group (Ramírez F. et.al., Nucleic Acids Res. 2014, 42(Web Server issue): W187-91). **b** Heatmap presenting the occupation ratio of histone markers' ChIP signals co-localized with hChe-RBPs' ChIP peaks at promoter and gene body regions, respectively. **c** Volcano plot showing the fold-change of H3K4me3 ChIP signals (left panel) and Pol II ChIP signals (right panel) on QKI5 commonly-activated genes upon knockdown of QKI5. The differential ChIP signals of H3K4me3 or Pol II between QKI5 knock-downed group and ctrl group were indentified by MAnorm software (Shao Z, et.al., Genome Biol. 2012, 13(3): R16).

Supported file 1-8 (for reviewer #1 major concern 3)

Supported file 1-8 a Percentage of CD14⁺/CD11b⁺ cells among HSPCs within Ctrl- or QKI5/QKI5 M1-overexpressing population and shCtrl- or shQKI5-treated population after 8 days of monocytic differentiation detected by flow cytometry. **b** Statistical result of flow cytometry results in (a). Error bars indicate standard deviations of three biological replicates. Asterisks indicate significant differences between the indicated samples (**P-value<0.01, ***P-value<0.001, ****P-value<0.0001, t test).

Supported file 1-9 (for reviewer #1 major concern 4)

a

b (Additional file 1: Figure S5c)

Supported file 1-9 a GO functional enrichment analysis of QKI5 ChIP-target genes. **b** GO functional enrichment analysis of the intersection of genes generated by QKI5 ChIP-targets genes and QKI5 commonly-activated genes.

Supported file 1-10 (for reviewer #1 major concern 5)

Supported file 1-10 a In vitro association of QKI5 with the CXCL2 promoter sequence identified by the DNA EMSA assay in which a 5'-biotin-labeled wild type and mutant CXCL2 promoter probes as well as ssDNA, DNA/RNA hybrid, and RNA probes with the same sequence of wild type CXCL2 promoter probe were used. The corresponding unlabeled (“cold”) probe was used in the competitive assay. **b** DNA EMSA assay using a 5'-biotin-labeled ChIP motif probe consisted of 2 tandem QKI5 ChIP motif repeats. The corresponding unlabeled (“cold”) probe was used in the competitive assay.

Supported file 2-1 (for reviewer #2 comments (1))

(Additional file 1: Figure S2b)

Supported file 2-1 Immuno-blots of hChe-RBPs immunoprecipitated using indicated antibodies. IgG was used as negative control of immunoprecipitation assay.

Supported file 2-2 (for reviewer #2 comments (1))

Supported file 2-2 a Immuno-blot detection of QKI5 in Ctrl- and QKI5-knockdown THP-1 cells. **b** Immuno-blot of QKI5 after QKI5 immunoprecipitation in Ctrl- and QKI5-knockdown THP-1 cells.* **c** Bar plot showing eCLIP-seq peak counts in Ctrl- and QKI5-knockdown THP-1 cells. The CLIP peaks were filtered by $P\text{-value} < 10e-7$ and fold change (IP/input) > 8 . **d** Pie chart presenting the overlapping or non-overlapping ratio of QKI5 CLIP peaks in Ctrl- and QKI5-knockdown THP-1 cells. **e** Boxplot of the enrichment levels of Ctrl CLIP-seq peaks overlapped or non-overlapped by QKI5-knockdown peaks. **f** Scatter plot showing the read counts covered by non-overlapped and overlapped peaks in Ctrl- and QKI5-knockdown CLIP-seq datasets. The reads counts were calculated by Deeptools software (Ramírez F. et.al., Nucleic Acids Res. 2014, 42(Web Server issue): W187-91). **g** IGV plots of QKI5 CLIP datasets generated from Ctrl- and QKI5-knockdown THP-1 cells at indicated locus. (* the blot in **a** was intercepted from blot in **b**)

Supported file 2-3 (for reviewer #2 comments (2))

a (Additional file 1: Figure S2c)

b

Supported file 2-3 a Repeatability test of ChIP-seq (left panel) and CLIP-seq (right panel) data. Pearson correlation coefficient was used to evaluate the repeatability of two biology replicates in ChIP-seq and CLIP-seq datasets. **b** IGV-plot of QKI5 CLIP-seq datasets on the indicated gene loci with 2 replicates.

Supported file 2-4 (for reviewer #2 comments (3))

b (Fig. 2j)

Supported file 2-4 a Heatmap showing the proportion of ChIP-seq/CLIP-seq overlapped peaks in each dataset (ChIP-seq: left panel; CLIP-seq peaks: right panel) of the hChe-RBPs. **b** Upper panel: Odds ratio of expression levels determined by RNA-seq upon hChe-RBP knocking down on hChe-RBP-occupied versus non-occupied genes (*P-value < 0.05, double-tail Fisher's exact test). The odds ratio of IgG ChIP-seq dataset versus RNA-seq datasets upon hChe-RBPs knocking down was used as a randomized control. Lower panel: Formula of odds ratio.

Second round of review

Reviewer 1

The authors have thoroughly addressed the comments i had to their manuscript. i do not have any further comments and I find the current version of the manuscript suitable for publication by Genome Biology.